 LongCat

# MemOCR: Layout-Aware Visual Memory for Efficient Long-Horizon Reasoning

Yaorui Shi [1 2 *]  Shugui Liu [1]  Yu Yang [2]  Wenyu Mao [1]  Yuxin Chen [3 2 *]
Qi Gu [2]  Hui Su [2]  Xunliang Cai [2]  Xiang Wang [1]  An Zhang [1]

## Abstract

Long-horizon agentic reasoning necessitates effectively compressing growing interaction histories into a limited context window. Most existing memory systems serialize history as text, where token-level cost is uniform and scales linearly with length, often spending scarce budget on low-value details. To this end, we introduce **MemOCR**, a multimodal memory agent that improves long-horizon reasoning under tight context budgets by allocating memory space with adaptive information density through visual layout. Concretely, MemOCR maintains a structured rich-text memory (*e.g.,* headings, highlights) and renders it into an image that the agent consults for memory access, visually prioritizing crucial evidence while aggressively compressing auxiliary details. To ensure robustness across varying memory budgets, we train MemOCR with reinforcement learning under budget-aware objectives that expose the agent to diverse compression levels. Across long-context multi-hop and single-hop question-answering benchmarks, MemOCR outperforms strong text-based baselines and achieves more effective context utilization under extreme budgets. Our code is available at https://github.com/meituan/MemOCR.

## 1. Introduction

The evolution of large language models (LLMs) has empowered autonomous agents to tackle complex, long-horizon tasks that necessitate robust long-horizon reasoning (Luo et al., 2025; Du et al., 2025b; Matarazzo & Torlone, 2025). However, as an agent accumulates extensive interaction history over its lifespan, the sheer volume of data inevitably overwhelms the hard constraints of the context window, creating a fundamental bottleneck (Vaswani et al., 2017; Hsieh et al., 2024; Modarressi et al., 2025). At the core of long-horizon reasoning is memory management under a finite working context: agents must continually decide what past information to store and what to retrieve into the context window (Fang et al., 2025; Hu et al., 2025; Zhang et al., 2025b). This essentially constitutes a budget allocation problem, where the objective is to maximize the density of task-relevant information within a limited number of tokens (*i.e.,* the memory budget) to support the current decision.

Leading approaches generally construct the working context using textual forms, which can be categorized into two paradigms. Early works populate the context by retrieving and injecting raw historical segments (*e.g.,* past chats) as uncompressed memory (Zhang et al., 2025a; Jin et al., 2025; Song et al., 2025). Specifically, the agent retrieves relevant passages and inserts the top-$k$ snippets to fill the working context, as demonstrated in Figure 1(a). While this preserves original details, the retrieved snippets can be redundant or noisy, diluting information density and potentially exhausting the context budget (Shi et al., 2025b; Wu et al., 2025). Instead of storing raw information, recent works (Yu et al., 2025; Wang et al., 2025; Chhikara et al., 2025; Shi et al., 2025a) compress past interactions into a compact textual summary, and maintain it via incremental updates or overwrites. In principle, summarization distills task-relevant information (*i.e.,* concepts crucial for answering future queries) from noisy history, providing a cleaner context to support decision-making.

However, the textual memory paradigm suffers from an intrinsic limitation: linear token scaling. Even though summarization alleviates redundancy, representing memory as text tightly couples storage cost to information content — retaining more auxiliary details or explanatory context inevitably requires proportionally more tokens (Feng et al., 2026; Fang et al., 2025; Sun et al., 2025). This coupling squanders the limited memory budget on non-critical supporting facts. As conceptually illustrated in Figure 1(b), text imposes uniform information density: to maintain 100 tokens of crucial information, the system is compelled to retain a substantial volume of auxiliary details (depicted as

*Work done during an internship at Meituan [1]University of Science and Technology of China, Hefei, China [2]Meituan, Beijing, China [3]National University of Singapore, School of Computing, Singapore. Correspondence to: An Zhang <an_zhang@ustc.edu.cn>, Qi Gu <guqi03@meituan.com>.

*Proceedings of the 43rd International Conference on Machine Learning*, Seoul, South Korea. PMLR 306, 2026. Copyright 2026 by the author(s).

∼900 tokens), lacking the flexibility to selectively downsample less important context. Recent visual-text compression methods explore rendering textual content as images to reduce token overhead in long-context scenarios (Wei et al., 2025; Cheng et al., 2025; Feng et al., 2026; Wang et al., 2026; Zhao et al., 2025). These approaches render text as into a visual format, without learned content selection or spatial allocation across the canvas.

We propose a paradigm shift from 1D textual memory to **2D visual memory**, representing history as an image rather than a token stream. The core benefit is **adaptive information density**: the agent can explicitly allocate the limited budget non-uniformly by controlling visual salience. Crucial evidence is rendered with prominent typography and high-visibility layout (*e.g.,* headers, bold, larger font), while auxiliary details are compressed into visually smaller text. This allows the agent to pack substantially more content into far fewer visual tokens while keeping key evidence readable under aggressive compression (Figure 1(c)). The overall budget can be controlled by resolution manipulation (*e.g.,* downsampling), providing a flexible budget-fidelity tradeoff without changing the memory content (Wei et al., 2025).

To this end, we introduce **MemOCR**, a multimodal memory agent that improves long-horizon reasoning under tight context budgets by allocating memory space with **adaptive information density** through visual layout. As shown in Figure 2, MemOCR formulates and utilizes visual memory with a two-stage memory lifecycle. *(1) Memory Drafting (Text Domain):* upon receiving new experience, the agent incrementally edits a persistent rich-text memory, including both updating content and assigning visual priority via structure and formatting. These explicit salience cues determine how memory components compete for limited space, enabling non-uniform budget allocation. *(2) Memory Reading (Vision Domain):* a lightweight renderer compiles the rich text into a 2D memory image, which becomes the agent's sole working context at query time. The agent then reads this memory image to produce an answer. We train MemOCR via RL with budget-aware training objectives that expose the agent to various memory budgets, forcing it to write crucial evidence highly visible under extreme budgets, while keeping auxiliary details in lower-priority regions.

We evaluate MemOCR on both multi-hop (*e.g.,* HotpotQA (Yang et al., 2018)) and single-hop (*e.g.,* Natural Questions (Kwiatkowski et al., 2019)) question-answering (QA) benchmarks across various context lengths and memory budgets (*cf.* §4.2). Across settings, MemOCR outperforms text-memory agents when the budget is sufficient, and exhibits substantially smaller performance drops as the budget tightens, yielding roughly an $8\times$ improvement in effective context utilization (*cf.* §4.3). Moreover, we find that visual salience is functionally important: weakening or removing

layout-based emphasis directly harms robustness under low budgets, and MemOCR learns to place more important information in more visually accessible regions (*cf.* §4.4). Finally, ablation studies verify the contribution of budget-aware training objectives (*cf.* §4.5). Complexity analysis in Appendix D.2 further reveals visual memory does not introduce much computational overhead.

## 2. Preliminaries

In this section, we formalize context management for long-horizon agent reasoning and pinpoint the uniform information density issue in text-based memory paradigms. See Appendix A for a discussion of related work.

### 2.1. Problem Formulation

We consider an LLM agent operating in a persistent environment, where information arrives sequentially as a discrete stream of text chunks $\mathcal{C} = \{C_1, C_2, \ldots, C_T\}$. At each step $t$, the agent receives a new observation $C_t$. Given a user question $Q$, the agent's goal is to produce an accurate answer $A$ based on the cumulative history $\mathcal{C}$.

**Raw History Memory.** Ideally, the agent would condition its generation on the full history $\mathcal{C}$:

$$A \sim \pi_\theta(\cdot \mid \mathcal{C}, Q), \tag{1}$$

where $\pi_\theta$ is the agent policy. However, as $T$ grows, the length of $\mathcal{C}$ increases linearly and eventually exceeds the effective attention window of the underlying LLM. This makes raw-history conditioning a suboptimal strategy for utilizing the finite context window.

**Textual Summary Memory.** A common remedy is to maintain a compressed textual summary memory state $M_t$ that serves as a surrogate for $\{C_i\}_{i=1}^t$. Given a question $Q$, the agent iteratively updates the memory $M_t$ to retain information that is most relevant to answering $Q$:

$$M_t \sim \pi_\theta(\cdot \mid M_{t-1}, C_t, Q), \quad t \in \{1, \ldots, T\}. \tag{2}$$

After processing all chunks, the agent produces the answer $A$ conditioned on the final memory:

$$A \sim \pi_\theta(\cdot \mid M_T, Q). \tag{3}$$

This formulation captures a query-conditioned summarization workflow: memory is refined over time with respect to $Q$, and the final answer $A$ is generated from the latest memory state $M_T$.

### 2.2. Memory Budget and Uniform Information Density

We denote the context budget by $\mathcal{B}$ (in tokens for text-based contexts), which constrains the working context available at

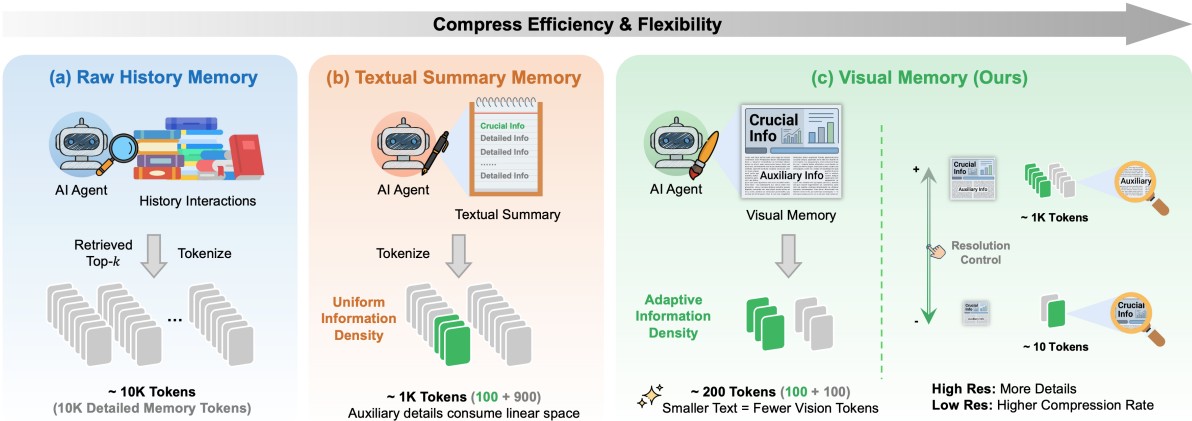

*Figure 1.* **Comparison of memory paradigms.** (a) **Raw History Memory** fetches relevant history passages but suffers from noise and redundancy. (b) **Textual Summary Memory** allows the agent to summarize the history but suffers from uniform information density, where auxiliary details (gray) consume as much token space as crucial information (green). (c) **Visual Memory (Ours)** allocates memory budget via visual layout to achieve adaptive information density.

inference time. For textual summary memory, the constraint is typically written as

$$|M_T| \leqslant \mathcal{B}. \tag{4}$$

We identify uniform information density as a key bottleneck of text-serialized memory. In text-based memory, every token occupies the same unit of budget regardless of semantic importance. Consequently, auxiliary details (*e.g.,* background and explanations) impose a rigid cost that competes directly with crucial evidence under the same token budget. This makes it difficult to allocate budget non-uniformly across memory components: keeping more supporting details inevitably reduces the space available for key evidence, even if those details are lower priority. This observation motivates a different representation in which context cost is not tied to word count, to enable explicit control over how different memory components consume the budget.

## 3. Method: MemOCR

In this section, we introduce MemOCR, a multimodal memory agent that improves long-horizon reasoning under tight context budgets by allocating memory space with adaptive information density through visual layout. As shown in Figure 2, MemOCR formulates and utilizes visual memory with a two-stage lifecycle: memory drafting in the text domain (§3.1), followed by memory reading in the vision domain (§3.2) after rendering. Finally, we present our budget-aware training objectives that train the agent to remain effective under different memory budgets (§3.3).

### 3.1. Memory Drafting in the Text Domain

The first stage corresponds to the text-domain drafting process (Figure 2(a)). Here, the agent functions as a memory

drafter that incrementally maintains a persistent rich-text memory, denoted as $M_t^{\text{RT}}$. Unlike plain-text summaries, rich text explicitly encodes visual priority via structure and formatting (*e.g.,* headings, indentation, bolding, font size), which later determines how different memory components compete for limited space on the canvas. Unless otherwise specified, we use Markdown as the carrier format.

At step $t$, given the previous memory state $M_{t-1}^{\text{RT}}$ and the new chunk $C_t$, the agent produces an updated memory:

$$M_t^{\text{RT}} \sim \pi_\theta(\cdot \mid M_{t-1}^{\text{RT}}, C_t), \quad t \in \{1, \dots, T\}. \tag{5}$$

The role of drafting is to decide what to keep and, crucially, what to emphasize: important evidence is assigned higher visual priority (*e.g.,* prominent headings or bold text), while auxiliary details are written in lower-priority regions. Importantly, the drafting process is budget-agnostic: the agent does not condition on the runtime memory budget when generating $M_t^{\text{RT}}$. Instead, it produces a single rich-text memory whose internal salience structure enables non-uniform budget allocation after rendering.

### 3.2. Memory Reading in the Vision Domain

The drafted rich-text memory is transformed into visual memory by a lightweight renderer $\mathcal{R}$ that bridges the text and vision domains:

$$V_T = \mathcal{R}(M_T^{\text{RT}}), \quad t \in \{1, \dots, T\}, \tag{6}$$

where $V_T$ is the rendered memory image at the final step. After rendering, memory cost is measured by the number of visual patch tokens rather than text length. In the memory image, layout and typography directly control information density: for a text segment of length $L$ rendered at font

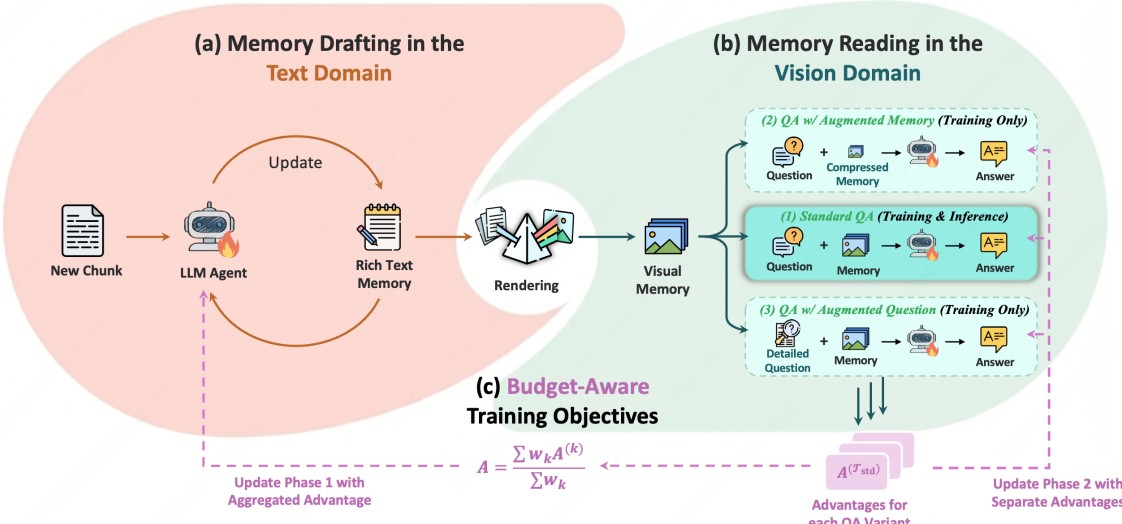

*Figure 2.* **Framework of MemOCR.** (a) **Memory Drafting (Text Domain):** The LLM agent incrementally updates a rich-text memory based on new incoming chunks, assigning visual priority via formatting and structure. (b) **Memory Reading (Vision Domain):** The rich text is rendered into a 2D memory image, which serves as the agent's sole working context for answering queries. (c) **Budget-Aware Training Objectives:** We train the agent under varying degrees of memory compression. The drafting ability is updated via aggregated advantages, while the reading ability is updated via separate advantages.

scale $s$, the occupied area (and thus approximate visual-token cost) scales as $\mathcal{O}(L \cdot s^2)$. Therefore, rendering crucial evidence with larger scale and placing it in high-visibility regions, while rendering auxiliary details with smaller scale, decouples semantic content from context cost. This realizes **adaptive information density**: crucial evidence remains readable under compression, while auxiliary details are compacted into lower-priority (and thus less visible) regions.

At query time, MemOCR functions as a memory reader in the vision domain (Figure 2(b)). The rendered image $V_T$ serves as the agent's working context, and the answer $A$ is generated by:

$$A \sim \pi_\theta(\cdot \mid V_T, Q). \tag{7}$$

To control the effective memory budget, resolution manipulation (*e.g.,* downsampling) can be applied to rendered image $V_T$ so that the resulting number of visual tokens does not exceed the budget.

### 3.3. Budget-Aware Training Objectives

Training MemOCR requires jointly optimizing drafting (text domain) and reading (vision domain). A major obstacle is a shortcut policy: without explicit constraints, the agent can place all information in a uniform, medium-sized style, making everything similarly visible on the canvas and bypassing the intended trade-off between crucial evidence and auxiliary details. This collapses adaptive information density back into uniform density.

To prevent this shortcut, we train MemOCR via Group Relative Policy Optimization (GRPO) (Shao et al., 2024) with

budget-aware training objectives based on data augmentation. As illustrated in Figure 3, we construct three complementary QA tasks for the same drafted memory:

**1. Standard QA ($\mathcal{T}_{\mathbf{std}}$).** We use the unmodified question from the training dataset and a memory budget of 512 tokens. The objective is to ensure global QA correctness when the visual memory is provided with sufficient tokens.

**2. QA w/ Augmented Memory ($\mathcal{T}_{\mathbf{augM}}$).** We deliberately downsample the rendered memory image by $4\times$ per dimension (*i.e.,* $16\times$ fewer pixels). Under severe compression, low-priority fine-grained details become illegible, while sufficiently prominent layout cues remain readable. This forces the drafter to assign enough visual priority to crucial evidence so that it survives resolution decay and remains retrievable under extreme budgets.

**3. QA w/ Augmented Question ($\mathcal{T}_{\mathbf{augQ}}$).** While crucial evidence must be salient, auxiliary details should not be discarded entirely. We therefore construct detail-oriented questions that target specific auxiliary information in the latest memory $M_T^{\mathrm{RT}}$, and provide the uncompressed visual memory. The objective encourages the agent to be able to identify low-priority fine-grained details when explicitly queried, given sufficient tokens.

**Optimization with Reinforcement Learning.** For each training instance, we sample a group of outputs and compute task-specific rewards and advantages for the three scenarios above. Following Figure 2(c), the **reading** behavior

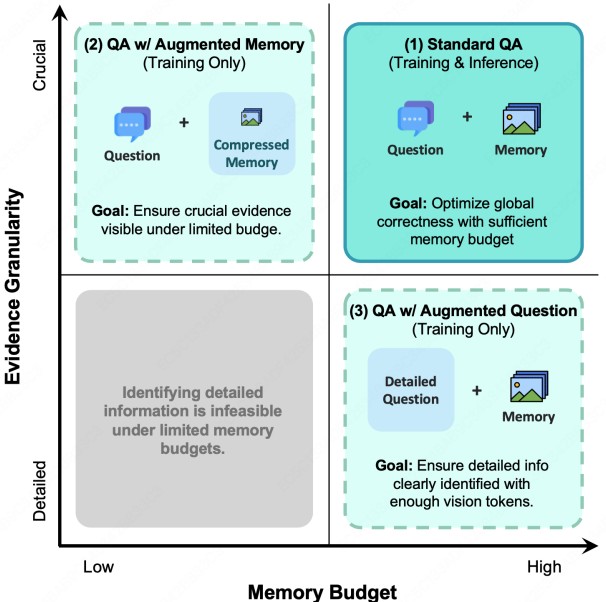

*Figure 3.* **Design of the budget-aware training objectives.** (1) **Standard QA** uses the unmodified question and memory for global correctness. (2) **QA w/ Augmented Memory** requires the visibility of crucial evidence even when the visual memory is heavily compressed. (3) **QA w/ Augmented Question** ensures detailed information is clearly identified with sufficient tokens. The low-budget, high-detail setting (gray area) is excluded as identifying detailed features under severe compression is infeasible.

is updated using separate task-specific advantages, since each scenario requires different visual reasoning behavior. In contrast, the **drafting** behavior must produce a single layout that serves all scenarios; therefore, it is updated via an aggregated advantage:

$$A = \frac{\sum_{k \in \mathcal{K}} w_k \cdot A^{(k)}}{\sum_{k \in \mathcal{K}} w_k}, \tag{8}$$

where $\mathcal{K} = \{\mathcal{T}_{\text{std}}, \mathcal{T}_{\text{augM}}, \mathcal{T}_{\text{augQ}}\}$ and $w_k$ are pre-defined task weights. By maximizing this global signal, MemOCR learns a layout strategy that keeps crucial evidence visible under extreme compression (via $\mathcal{T}_{\text{augM}}$), with the ability to recover detailed information when budget permits (via $\mathcal{T}_{\text{augQ}}$).

# 4. Experiments

We evaluate MemOCR under long-horizon reasoning scenarios to answer four research questions (RQs):

**RQ1:** Does MemOCR improve overall QA performance under long contexts and varying budgets?

**RQ2:** Does layout control improve MemOCR's robustness under tight memory budgets?

**RQ3:** Does layout induce region-wise robustness under compression, and does MemOCR exploit it to realize adaptive information density?

**RQ4:** How do budget-aware training objectives contribute to the learned behavior?

## 4.1. Experimental Setup

**Datasets.** We train on HotpotQA (Yang et al., 2018) and pad each sample with distractor documents to reach ∼30K tokens during training. We evaluate on multi-hop HotpotQA and 2WikiMultiHopQA (2Wiki) (Ho et al., 2020), as well as single-hop Natural Questions (NQ) (Kwiatkowski et al., 2019) and TriviaQA (Joshi et al., 2017). During evaluation, contexts are padded to 10K/30K/100K tokens. We report subword exact match as accuracy, averaged over three runs.

**Baselines.** We compare MemOCR against two categories of baselines: (1) **Raw History Memory** with uncompressed context, including the standard Qwen2.5-Instruct (Yang et al., 2024), the Qwen model distilled from DeepSeek-R1 (R1-Distill Qwen) (DeepSeek-AI et al., 2025) and Qwen2.5-1M-Instruct (Yang et al., 2025); (2) **Textual Summary Memory**, represented by Mem0 (Chhikara et al., 2025), Mem-$\alpha$ (Wang et al., 2025) and MemAgent (Yu et al., 2025). We use Qwen2.5-7B-Instruct (Yang et al., 2024) as the default backbone LLM for text-based methods, and Qwen2.5-VL-7B-Instruct (Bai et al., 2025) for MemOCR.

**Memory Budget.** We study long-context QA with an explicit memory budget constraint $\mathcal{B} \in \{16, 64, 256, 1024\}$, where the "memory budget" controls the number of tokens occupied in the context window at answer time. For textual summary agents, we constrain summary length by only keeping the first $\mathcal{B}$ tokens in the latest memory state $|M_T|$. For MemOCR, we adjust the resolution of the rendered memory image so that the number of visual patch tokens is no greater than $\mathcal{B}$. Additional details are in Appendix C.

## 4.2. Overall Performance (RQ1)

We compare MemOCR with baselines across datasets, context lengths (10K/30K/100K), and budgets, to test whether MemOCR improves overall performance and remains robust as the budget tightens. Table 1 reports the main results.

**Obs. 1.1: MemOCR achieves the best overall performance across different context lengths.** MemOCR attains the highest average accuracy from 10K to 100K contexts, indicating that MemOCR scales to long histories. For example, at 10K context with full budget, MemOCR reaches 74.6% average accuracy, surpassing the strongest baseline at 67.8%. Statistical significance is provided in Appendix D.1. We also observe poor performance on HotpotQA under 30K and 100K context lengths, and our analysis on bad cases helps explain this phenomenon in Appendix E.

 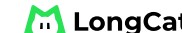 

*Table 1.* Comparison of accuracy (%) across different context lengths (10K, 30K, and 100K tokens). Best results are highlighted in **bold** and second best in underline. The percentages in the Average column represent the performance drop relative to the 1024-token budget.

| Method | Memory Budget (Tokens) | HotpotQA 10K | 30K | 100K | 2Wiki 10K | 30K | 100K | NQ 10K | 30K | 100K | TriviaQA 10K | 30K | 100K | Average 10K | 30K | 100K |
|---|---|---|---|---|---|---|---|---|---|---|---|---|---|---|---|---|
| *Raw History Memory* | | | | | | | | | | | | | | | | |
| Qwen2.5 | 100K | 70.3 | - | - | 53.9 | 61.1 | - | 50.8 | - | - | 71.1 | - | - | 61.5 | - | - |
| R1-Distill Qwen | 100K | 50.8 | 12.5 | 0.8 | 50.8 | 26.6 | 2.3 | 35.2 | 3.9 | 1.6 | 54.7 | 11.7 | 0.8 | 47.9 | 13.7 | 1.4 |
| Qwen2.5-1M | 100K | 75.0 | 66.4 | 68.8 | 67.2 | 65.6 | 54.7 | 53.1 | 43.8 | 50.8 | 71.1 | 77.3 | 59.4 | 66.6 | 63.3 | 58.4 |
| *Textual Summary Memory* | | | | | | | | | | | | | | | | |
| Mem0 | 1024 | 70.3 | 66.1 | 64.1 | 60.4 | 47.7 | 40.9 | 47.7 | 48.8 | 48.2 | 69.5 | 70.6 | 61.4 | 62.0 | 58.3 | 53.6 |
| | 256 | 71.1 | 65.6 | 63.8 | 57.8 | 44.8 | 43.5 | 46.9 | **51.1** | 49.7 | 69.5 | 70.6 | 62.8 | 61.3 (−1.1%) | 58.0 (−0.5%) | 55.0 (+2.5%) |
| | 64 | 56.7 | 60.9 | 56.2 | 53.9 | 42.2 | 43.3 | 46.9 | 43.8 | 50.0 | 66.4 | 65.9 | 62.5 | 56.0 (−9.7%) | 53.2 (−8.8%) | 53.0 (−1.2%) |
| | 16 | 37.0 | 43.3 | 34.9 | 28.4 | 31.5 | 27.6 | 41.4 | 34.1 | 43.0 | 65.3 | 62.8 | 58.9 | 43.0 (−30.6%) | 42.9 (−26.4%) | 41.1 (−23.4%) |
| Mem-α | 1024 | 75.0 | 77.9 | 79.4 | 47.7 | 40.1 | 50.0 | 45.6 | 44.8 | 43.0 | 21.1 | 23.2 | 19.3 | 47.3 | 46.5 | 47.9 |
| | 256 | 57.8 | 57.3 | 57.0 | 38.3 | 32.8 | 41.7 | 47.9 | 33.6 | 38.0 | 17.2 | 24.2 | 18.3 | 40.3 (−14.9%) | 37.0 (−20.5%) | 38.7 (−19.2%) |
| | 64 | 25.8 | 29.7 | 28.9 | 34.9 | 31.5 | 39.1 | 27.1 | 25.0 | 24.0 | 13.0 | 19.5 | 13.8 | 25.2 (−46.8%) | 26.4 (−43.2%) | 26.4 (−44.9%) |
| | 16 | 24.0 | 30.2 | 26.3 | 35.4 | 32.3 | 40.1 | 23.7 | 26.3 | 24.2 | 11.7 | 19.0 | 14.3 | 23.7 (−50.0%) | 27.0 (−42.0%) | 26.2 (−45.3%) |
| MemAgent | 1024 | 82.3 | **78.9** | 79.4 | 65.4 | 63.8 | 61.2 | 53.4 | 46.1 | 55.5 | 70.1 | 78.1 | 66.7 | 67.8 | 66.7 | 65.7 |
| | 256 | 82.3 | 76.8 | 76.6 | 66.1 | 62.5 | 58.1 | 51.3 | 44.3 | 53.6 | 69.5 | 77.6 | 67.2 | 67.3 (−0.7%) | 65.3 (−2.1%) | 63.9 (−2.8%) |
| | 64 | 50.8 | 52.3 | 51.0 | 38.5 | 42.7 | 36.2 | 48.7 | 36.5 | 47.9 | 64.6 | 69.5 | 59.6 | 50.7 (−25.3%) | 50.3 (−24.7%) | 48.7 (−25.9%) |
| | 16 | 24.0 | 26.6 | 23.2 | 28.4 | 26.3 | 16.7 | 24.5 | 20.1 | 27.3 | 49.7 | 56.3 | 41.7 | 31.6 (−53.3%) | 32.3 (−51.6%) | 27.2 (−58.6%) |
| *Visual Memory* | | | | | | | | | | | | | | | | |
| **MemOCR (Ours)** | 1024 | **84.8** | 75.1 | 78.3 | **72.2** | **73.7** | 62.7 | **61.8** | 49.2 | 55.4 | 79.6 | **81.3** | 69.8 | **74.6** | **69.8** | 66.6 |
| | 256 | 82.2 | 75.4 | 75.7 | 71.2 | 72.8 | **65.5** | 57.3 | 48.8 | **58.3** | 79.7 | 80.9 | 68.9 | 72.6 (−2.7%) | 69.5 (−0.5%) | 67.1 (+0.8%) |
| | 64 | 77.6 | 68.1 | 67.2 | 62.9 | 66.2 | 62.4 | 51.0 | 43.6 | 47.5 | 77.6 | 80.7 | 70.2 | 67.3 (−9.8%) | 64.7 (−7.4%) | 60.7 (−8.8%) |
| | 16 | 67.2 | 57.9 | 52.4 | 57.9 | 56.0 | 45.7 | 42.8 | 35.9 | 45.9 | **80.8** | 72.2 | 67.2 | 62.2 (−16.6%) | 55.5 (−20.5%) | 52.8 (−20.7%) |

**Obs. 1.2: MemOCR degrades more gracefully under tight budgets.** Textual summary methods suffer catastrophic degradation as the budget tightens. For instance, on 10K contexts, MemAgent drops from 67.8% (1024 tokens) to 31.6% (16 tokens) on average. In contrast, MemOCR preserves 62.2% average accuracy at 16 tokens, corresponding to only a 16.6% relative drop. These results suggest that MemOCR retains task-relevant evidence more effectively under severe budget constraints.

**Obs. 1.3: On single-hop tasks, low memory budget can be sufficient for sparse evidence.** On NQ and TriviaQA, tightening the budget does not necessarily reduce accuracy. For example, on TriviaQA (10K context), MemOCR achieves 80.8% at 16 tokens, even higher than 79.6% at 1024 tokens. We attribute this to single-hop questions relying on atomic evidence, where a lower-resolution memory can still preserve the critical cues while filtering background noise. This phenomenon is not observed for textual baselines, whose performance decreases with smaller budgets.

**Comparison against RAG-based Memory.** We further compare MemOCR against recent RAG-based memory methods (Fang et al., 2025; Liu et al., 2026; Zhao et al., 2026) that maintain unlimited external storage and retrieve relevant passages at inference time. Table 2 reports results under 10K and 30K contexts.

**Obs. 1.4: MemOCR consistently outperforms RAG-based memory methods that operate with unlimited storage budgets.** At 10K context, MemOCR with $B =$

1024 achieves 74.6% average accuracy, surpassing the best unlimited-budget baseline HyMem (56.4%) by 18.2 points. Even under an aggressive 64-token budget, MemOCR obtains 67.3% and exceeds all unlimited baselines by a wide margin. The gap persists at 30K context where MemOCR with $B = 256$ reaches 69.5% versus 59.0% for the strongest competitor. These results demonstrate that learned memory compression with adaptive information density is more effective than retrieving from large uncompressed stores.

**Token Fairness and Generalization.** A natural question is whether MemOCR's gains stem from the vision encoder's higher information density, rather than the learned layout policy. We conduct three comparisons: (a) *w.r.t.* budget fairness, where text-based agents are granted 4× more tokens; (b) use summarization instead of truncation in textual memory agents; and (c) zero-shot experiments on LOCOMO (Maharana et al., 2024). The results are in Table 3.

**Obs. 1.4: MemOCR outperforms text baselines with 4× more token budgets.** We compare MemOCR with $B = N$ against baselines with $B = 4N$ (Table 3a). MemOCR still yields superior or comparable performance comparing against textual memory baselines with 4× budgets. At the most extreme setting, MemOCR achieves 62.2% at $B = 16$, which outperforms MemAgent with $B = 64$ by 11.5 points.

**Obs. 1.5: MemOCR outperforms text baselines augmented with LLM summarization.** According to Table 3b, replacing naive truncation with summarization improves MemAgent, *e.g.,* from 31.6 to 58.5 on $B = 16$.

*Table 2.* Comparison with recent RAG-based memory baselines with unlimited storage (accuracy %).

| Method | Memory Budget | HotpotQA | | 2Wiki | | NQ | | TriviaQA | | Average | |
| | (Tokens) | 10K | 30K | 10K | 30K | 10K | 30K | 10K | 30K | 10K | 30K |
|---|---|---|---|---|---|---|---|---|---|---|---|
| SimpleMem | ∞ | 73.4 | 67.2 | 52.3 | 41.4 | 38.3 | 37.5 | 57.8 | 62.5 | 55.5 | 52.2 |
| LightMem | ∞ | 64.1 | 66.4 | 59.4 | 50.8 | 50.0 | 46.1 | 42.2 | 38.3 | 53.9 | 50.4 |
| HyMem | ∞ | 65.3 | 70.0 | 45.8 | 49.7 | 45.3 | 44.5 | 69.2 | 71.6 | 56.4 | 59.0 |
| HyMem | 256 | 66.1 | 68.4 | 41.9 | 48.9 | 43.8 | 43.8 | 69.2 | 70.8 | 55.3 | 58.0 |
| | 1024 | **84.8** | 75.1 | **72.2** | 73.7 | **61.8** | 49.2 | 79.6 | 81.3 | **74.6** | 69.8 |
| MemOCR | 256 | 82.2 | **75.4** | 71.2 | 72.8 | 57.3 | 48.8 | 79.7 | 80.9 | 72.6 | 69.5 |
| | 64 | 77.6 | 68.1 | 62.9 | 66.2 | 51.0 | 43.6 | 77.6 | 80.7 | 67.3 | 64.7 |

*Table 3.* Fairness and generalization experiments (accuracy %, 10K context). (a) Budget fairness (MemOCR w/ $B = N$ vs text baselines w/ $B = 4N$, Avg over 4 benchmarks). (b) Text summarization replacing truncation for MemAgent (Yu et al., 2025) (Avg over 4 benchmarks. "+Summ." indicates replacing truncation with summarization). (c) LOCOMO (Maharana et al., 2024) zero-shot evaluation.

| | **(a) Budget Fairness** | | | **(b) vs Text Summarization** | | | **(c) LOCOMO Zero-Shot** | | |
| $N$ | MemOCR ($B = N$) | MemAgent ($B = 4N$) | Mem-$\alpha$ ($B = 4N$) | Budget | MemOCR | MemAgent+Summ. | Budget | Mem-$\alpha$ | MemOCR |
|---|---|---|---|---|---|---|---|---|---|
| 256 | **72.6** | 67.8 | 47.3 | 1024 | **74.6** | 66.8 | 1024 | 23.83 | **33.14** |
| 64 | **67.3** | 67.3 | 40.3 | 256 | **72.6** | 67.2 | 256 | 23.05 | **31.71** |
| 16 | **62.2** | 50.7 | 25.2 | 64 | **67.3** | 62.0 | 64 | 8.59 | **17.52** |
| | | | | 16 | **62.2** | 58.5 | 16 | 3.12 | **15.63** |

Under this setting, MemOCR still leads across all budgets. This demonstrates that MemOCR's advantage is not an artifact of comparing against poorly truncated text, but reflects a genuine benefit of the visual memory paradigm over optimized text compression.

**Obs. 1.6: MemOCR generalizes to zero-shot conversational memory.** On the LOCOMO (Maharana et al., 2024) long-term conversational memory benchmark (Table 3c), MemOCR outperforms Mem-$\alpha$ acrosss all budgets. These results confirm that the learned layout policy transfers beyond QA to conversational content.

### 4.3. Analysis on Visual Robustness (RQ2)

To identify the source of MemOCR's low-budget robustness, we compare against (i) *MemOCR w/o Visual Layout*, which preserves the visual modality but removes all formatting cues, and (ii) all three textual baselines on HotpotQA with a 10K context. Figure 4 reports both accuracy and relative drop from the 1024-token setting.

**Obs. 2.1: Visual layout significantly enhances low-budget robustness.** As the budget tightens, MemOCR degrades most gracefully, with substantially smaller drops than all baselines. Removing visual layout causes a marked additional drop, especially as the memory budget goes down. This additional drop indicates that MemOCR's robustness primarily comes from the layout-guided allocation of memory budgets, rather than the visual modality alone.

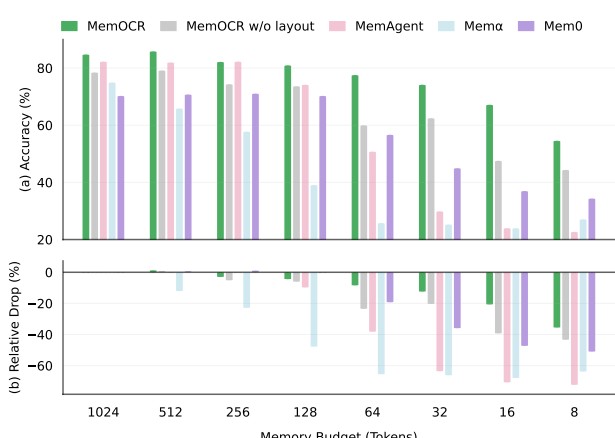

*Figure 4.* **Comparison of accuracy and relative performance drop across varying memory budgets (RQ2).** MemOCR degrades more gracefully than textual baselines as budgets tighten. Without visual layout, MemOCR's low-budget robustness drops significantly, which suggests that adaptive information density facilitates more efficient memory budget utilization.

**Obs. 2.2: MemOCR achieves an $8\times$ token-efficiency gain at extreme budgets.** At 8 tokens, MemOCR attains comparable accuracy to the baselines at 64 tokens, corresponding to an $8\times$ reduction in memory tokens (64→8) for similar performance.

### 4.4. Mechanism Verification (RQ3)

To answer RQ3, we verify a two-step mechanism of layout control: (1) *region-wise robustness*—different layout regions are not equally robust under visual compression, and

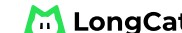

*Table 4.* Ablation study by progressively removing training objectives from MemOCR. Vanilla MemOCR uses $\mathcal{T}_{\text{std}}$, $\mathcal{T}_{\text{augM}}$, $\mathcal{T}_{\text{augQ}}$.

| Method | Memory Budget (Tokens) | HotpotQA 10K | 30K | 100K | 2Wiki 10K | 30K | 100K | NQ 10K | 30K | 100K | TriviaQA 10K | 30K | 100K | Average 10K | 30K | 100K |
|---|---|---|---|---|---|---|---|---|---|---|---|---|---|---|---|---|
| **MemOCR** | 1024 | **84.8** | 75.1 | **78.3** | **72.2** | **73.7** | 62.7 | **61.8** | **49.2** | 55.4 | 79.6 | **81.3** | 69.8 | **74.6** | **69.8** | 66.6 |
| | 256 | 82.2 | **75.4** | 75.7 | 71.2 | 72.8 | **65.5** | 57.3 | 48.8 | **58.3** | 79.7 | 80.9 | 68.9 | 72.6 (−2.7%) | 69.5 (−0.5%) | **67.1** (+0.8%) |
| | 64 | 77.6 | 68.1 | 67.2 | 62.9 | 66.2 | 62.4 | 51.0 | 43.6 | 65.8 | 77.6 | 80.7 | 65.8 | 67.3 (−9.8%) | 64.7 (−7.4%) | 60.7 (−8.8%) |
| | 16 | 67.2 | 57.9 | 52.4 | 57.9 | 56.0 | 45.7 | 42.8 | 35.9 | 45.9 | **80.8** | 72.2 | **67.2** | 62.2 (−16.6%) | 55.5 (−20.5%) | 52.8 (−20.7%) |
| w/o $\mathcal{T}_{\text{augM}}$ | 1024 | 75.2 | 71.7 | 68.9 | 53.4 | 55.5 | 51.1 | 49.2 | 46.5 | 47.8 | 67.6 | 74.2 | 57.0 | 61.3 | 62.0 | 56.2 |
| | 256 | 67.3 | 65.8 | 63.7 | 51.4 | 53.0 | 49.7 | 47.8 | 44.2 | 47.0 | 66.6 | 75.1 | 57.9 | 58.3 (−5.0%) | 59.5 (−3.9%) | 54.6 (−2.9%) |
| | 64 | 57.7 | 52.2 | 50.8 | 38.6 | 41.7 | 39.0 | 35.5 | 35.0 | 36.7 | 62.1 | 66.3 | 55.3 | 48.5 (−21.0%) | 48.8 (−21.2%) | 45.4 (−19.2%) |
| | 16 | 43.7 | 40.9 | 39.8 | 37.1 | 34.8 | 27.4 | 29.2 | 24.4 | 31.3 | 54.9 | 59.4 | 45.9 | 41.2 (−32.8%) | 39.9 (−35.6%) | 36.1 (−35.8%) |
| w/o $\mathcal{T}_{\text{augM}}$, $\mathcal{T}_{\text{augQ}}$ | 1024 | 74.7 | 68.5 | 71.1 | 53.9 | 57.7 | 47.8 | 50.8 | 38.8 | 42.3 | 69.3 | 72.5 | 63.6 | 62.2 | 59.4 | 56.2 |
| | 256 | 70.6 | 67.5 | 64.3 | 54.2 | 53.0 | 44.3 | 42.9 | 39.8 | 48.7 | 67.7 | 69.7 | 58.9 | 58.8 (−5.4%) | 57.5 (−3.2%) | 54.0 (−3.9%) |
| | 64 | 44.6 | 39.0 | 46.9 | 40.7 | 39.6 | 32.9 | 29.0 | 22.4 | 29.6 | 50.1 | 57.0 | 46.0 | 41.1 (−33.9%) | 39.5 (−33.5%) | 38.9 (−30.9%) |
| | 16 | 28.4 | 28.8 | 27.8 | 33.8 | 37.0 | 27.4 | 24.9 | 17.9 | 24.7 | 44.5 | 46.3 | 40.9 | 32.9 (−47.1%) | 32.5 (−45.2%) | 30.2 (−46.3%) |
| w/o $\mathcal{T}_{\text{std}}$, $\mathcal{T}_{\text{augM}}$, $\mathcal{T}_{\text{augQ}}$ | 1024 | 67.2 | 54.7 | 41.4 | 51.6 | 45.3 | 38.3 | 45.3 | 40.6 | 43.8 | 62.5 | 68.0 | 47.7 | 56.6 | 52.1 | 42.8 |
| | 256 | 62.5 | 51.6 | 46.9 | 46.9 | 46.9 | 31.3 | 46.9 | 36.7 | 42.2 | 71.1 | 70.3 | 58.6 | 56.8 (+0.3%) | 51.4 (−1.5%) | 44.7 (+4.6%) |
| | 64 | 62.5 | 50.8 | 42.2 | 47.7 | 43.0 | 32.8 | 46.9 | 36.7 | 39.8 | 66.4 | 67.2 | 53.1 | 55.9 (−1.4%) | 49.4 (−5.2%) | 42.0 (−1.8%) |
| | 16 | 33.6 | 30.5 | 28.9 | 34.4 | 25.0 | 28.1 | 27.3 | 25.0 | 28.1 | 54.7 | 60.2 | 46.9 | 37.5 (−33.8%) | 35.2 (−32.6%) | 33.0 (−22.8%) |

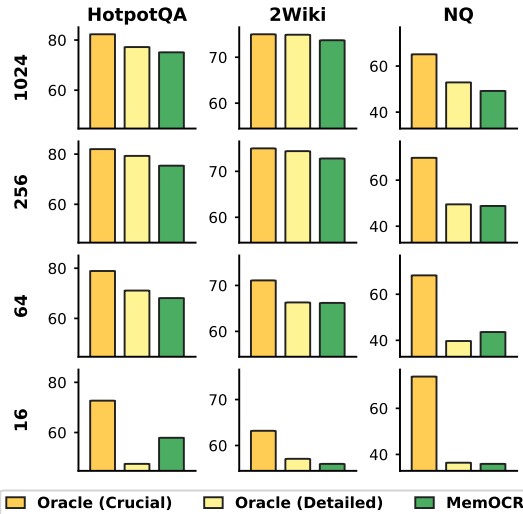

*Figure 5.* **Oracle analysis of layout regions (RQ3).** We compare MemOCR with oracle variants that inject ground-truth evidence into either the *Crucial* or *Detailed* region of the rendered memory. While both injections improve accuracy, injecting into the crucial region yields larger gains, especially under tight memory budgets.

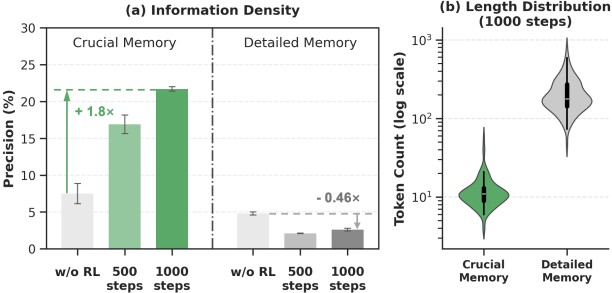

*Figure 6.* **RL induces adaptive information density (RQ3).** (a) With training, ground-truth evidence becomes more concentrated in the crucial region while decreasing in the detailed region. (b) The crucial region remains much shorter than the detail part.

regions is more likely to be lost under compression. In some extreme-budget cases (*e.g.,* HotpotQA at 16 tokens), injection can be harmful because the added content increases the memory image size, making it less legible overall.

**Obs. 3.2: RL enables adaptive information density.** Without RL, crucial and detailed regions have similar evidence density. As shown in Figure 6(a), MemOCR shifts precise evidence into the crucial region (precision $\uparrow \sim 1.8\times$) and reduces density in the detailed region (precision $\downarrow \sim 0.46\times$) during RL training. Meanwhile, the crucial region stays orders of magnitude shorter (Figure 6(b)). This demonstrates adaptive information density: key evidence is compactly preserved in visually high-priority areas, while details allocated to lower-priority regions.

### 4.5. Ablation Study over Training Objectives (RQ4)

We ablate MemOCR by progressively removing training objectives. Specifically, we compare vanilla MemOCR trained with full objectives ($\mathcal{T}_{\text{std}}+\mathcal{T}_{\text{augM}}+\mathcal{T}_{\text{augQ}}$) with three variants: (1) w/o $\mathcal{T}_{\text{augM}}$ (2) w/o $\mathcal{T}_{\text{augM}}$, $\mathcal{T}_{\text{augQ}}$, and (3) w/o $\mathcal{T}_{\text{std}}$, $\mathcal{T}_{\text{augM}}$, $\mathcal{T}_{\text{augQ}}$. Results are reported in Table 4.

(2) *evidence placement*—MemOCR exploits this asymmetry and stores important evidence into more visible regions. We evaluate both under the 30K-context setting using oracle injections (Figure 5) and memory statistics (Figure 6).

**Obs. 3.1: Evidence is more compression-robust in more visible regions.** We construct two oracle controls by injecting the same ground-truth evidence into different regions of MemOCR memories: *Oracle (Crucial)* inserts it into the crucial region (H1 headers), whereas *Oracle (Detailed)* inserts it into the detailed region (plain body text). As shown in Figure 5, Oracle (Crucial) consistently outperforms Oracle (Detailed), and the advantage grows as the budget tightens, indicating that information placed in lower-visibility

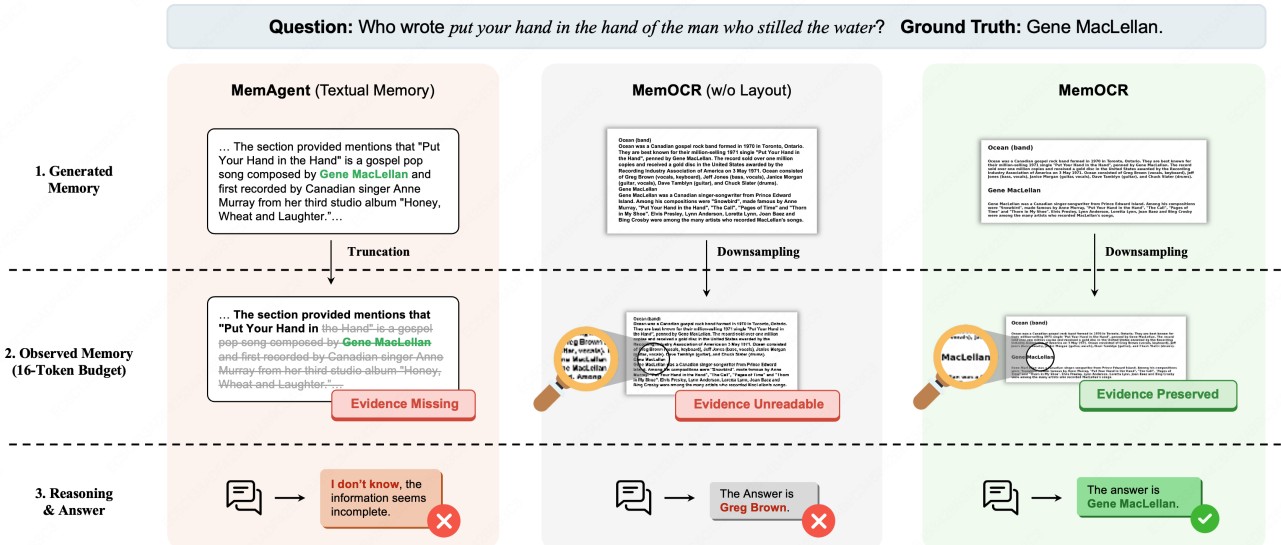

*Figure 7.* **Case study at an extreme memory budget (16 tokens).** (Left) The textual baseline fails due to hard truncation of the context. (Middle) MemOCR without layout control fails because uniform text becomes unreadable after down-sampling. (Right) MemOCR preserves the crucial evidence "Gene MacLellan" through adaptive layout, enabling correct reasoning even at low resolution.

**Obs. 4.1: Training is necessary for budget-robust memory layout.** Across datasets and context lengths, simply removing all the training signals underperforms substantially. This indicates that the desired layout arrangement behavior does not emerge reliably without explicit training.

**Obs. 4.2: Robustness gains accumulate with multiple training signals.** Using $\mathcal{T}_{std}$ alone yields the weakest trained variant, which reflects limited ability to prioritize question-relevant evidence through layout control. Adding $\mathcal{T}_{augQ}$ improves low-budget robustness, suggesting that the diverse queries strengthen evidence utilization. Further adding $\mathcal{T}_{augM}$ brings a larger boost in the low-budget regime, consistent with the concept of learning more effective layout control and memory budget allocation.

### 4.6. Case Study

To provide an intuitive illustration of the robustness trends in RQ2 (§4.3), we visualize the memory states of three agents answering the question *"Who wrote put your hand in the hand...?"* under an extreme 16-token budget (Figure 7).

**Textual summary memory fails under extreme budgets due to truncation.** For the textual baseline (MemAgent), the 16-token limit necessitates hard truncation of the memory. As a result, the critical entity "Gene MacLellan" is removed from the context window, leaving insufficient evidence for correct answering.

**MemOCR w/o Layout fails because uniform rendering becomes unreadable after downsampling.** As shown in

the middle panel of Figure 7, since the visual layout control via rich-text grammar is disabled, the memory image is rendered as a dense uniform text block without priority. When downsampled to 16 visual tokens (approximately 12K pixels in Qwen2.5-VL), the resolution becomes too low to resolve characters on the memory image, ultimately leading to an incorrect answer (*i.e.,* "Greg Brown").

**MemOCR succeeds by keeping crucial evidence legible via adaptive layout.** As shown in the rightmost panel of Figure 7, the agent isolates the crucial information ("Ocean Band" and "Gene MacLellan") into visually prominent regions. Even after aggressive downsampling blurs surrounding auxiliary text, the pixels corresponding to the crucial evidence remain recognizable, enabling correct answering.

## 5. Conclusion and Future Work

In this paper, we shift agentic memory management from linear textual streams to a flexible, spatial 2D canvas, termed visual memory. Building on this concept, we propose MemOCR, which dynamically manipulates visual layout and resolution to decouple information density from token cost, yielding strong robustness under extreme budget constraints. We discuss the limitations of MemOCR in Appendix B.

For future work, a natural next step is to generalize visual memory beyond QA to broader long-horizon agent settings, such as planning and tool-augmented reasoning, and to study long-term stability under lifelong updates. We also plan to improve the budget allocation policy by introducing more flexible rich-text formats (*e.g.,* HTML).

## Impact Statement

This paper aims to advance long-horizon agentic reasoning by introducing a visual-memory paradigm that represents interaction history on a 2D canvas and allocates limited context budgets via adaptive visual compression. By improving effective context utilization, our approach may enable more long-horizon multimodal agents.

At the same time, stronger long-horizon memory can amplify existing ethical and societal risks. First, agentic memory may increase privacy and security concerns if sensitive user information is stored, rendered, or inadvertently exposed through model outputs or logs. Second, the visual memory system may encourage harms in high-stakes domains since many of modern LLM-safety solutions are designed for text-only interactions. Third, visual rendering and OCR-style reading may introduce new failure modes, which could lead to hallucinated responses if the system cannot reliably recover key evidence.

These risks are not unique to our method but can be amplified by improved memory capacity. Mitigations include adopting strict data retention policies, obtaining user consent and increasing transparency of the memory system with access control and encryption, and evaluating robustness and bias across demographics and domains.

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

# A. Related Work

**Reinforcement Learning in LLM Agents.** In recent years, reinforcement learning (RL) (Kaelbling et al., 1996) has emerged as a powerful paradigm for post-training large language models (LLMs). While initial efforts focused on human preferences (Ouyang et al., 2022) or distilled reward models (Bai et al., 2022), the field has gradually shifted toward rule-based feedback, demonstrating great potential in enhancing model capabilities. Key algorithmic contributions include proximal policy optimization (Schulman et al., 2017), based on generalized advantage estimation (Schulman et al., 2015), and GRPO (Shao et al., 2024), which utilizes group normalization, improving the optimization stability. Building upon these foundations, recent research has extended RL to optimization of complex and long-horizon trajectory, specifically targeting multi-turn interactions with tool utilization (Jin et al., 2025; Shi et al., 2025b; Xue et al., 2025).

**Agentic Memory Management.** Given the context window limits of LLMs, a memory mechanism is essential for agents to retain information for long-horizon reasoning. Mainstream approaches typically employ textual representations, generally falling into two paradigms. The first treats raw historical segments as memory and directly injects them into the working context (Jin et al., 2025; Song et al., 2025; Shi et al., 2025b; Du et al., 2025a; Li et al., 2025). For example, Search-r1 (Jin et al., 2025) enables multi-turn searching by leveraging raw history throughout the reasoning process. The second paradigm adopts textual summary memory, where long-context information is compressed into concise text forms rather than retaining the full raw history (Duverger et al., 2024; Yu et al., 2025; Wang et al., 2025; Bian et al., 2026; Hu et al., 2026; Shi et al., 2025a; Sheng et al., 2026). This approach distills historical interactions into summaries, maintained via dynamic updates or overwrites. For instance, MemAgent (Yu et al., 2025) proposes a "memorizing while reading" paradigm, summarizing and distilling task-relevant information step by step. MemAlpha (Wang et al., 2025) trains the agent to manage complex hierarchical memory systems, *i.e.,* to extract, store, and update memory corpora of varying sizes and importance.

**OCR for Context Compression.** Optical Character Recognition (OCR) (Arlazarov et al., 2022; Smith, 2007) is a well-established technology that is widely utilized for extracting textual content embedded in image format. In recent research advances, OCR has beed explored as an innovative vision-text compression paradigm (Wei et al., 2025; Xing et al., 2025; Cheng et al., 2025). Unlike the conventional practice of directly inputting long context into LLMs, this OCR-driven method encodes long context into visual representations, *e.g.,* VTC-R1 (Wang et al., 2026; Zhao et al., 2025). By leveraging the high information density of visual tokens to reduce token overhead, this approach holds the potential to revolutionize the memory architecture of agents like AgentOCR (Feng et al., 2026).

# B. Limitations

While MemOCR demonstrates strong token-efficiency and robustness under tight budgets, it has several limitations.

- **Dependence on vision/OCR robustness.** MemOCR relies on the backbone vision-language model to accurately read heavily downsampled text and layout cues. Failures in visual perception (e.g., blur, small fonts, rendering flaws) can directly translate into missing or hallucinated evidence, especially at extreme budgets.

- **Layout policy may be task-specific.** The learned salience allocation is optimized for long-context QA-style supervision and the training distributions. It may not transfer optimally to other agentic workloads (*e.g.,* planning, tool-use logs, dialog personalization) where the notion of "crucial evidence" differs or evolves over time.

- **Additional computational overheads.** Although rendering is lightweight compared to model inference, the vision language modeling introduces extra latency and complexity in real deployments, especially with a relatively small context length (*e.g.,* the 10K context length in Figure 11).

# C. Implementation Details

This appendix provides the engineering details required to reproduce MemOCR, including the end-to-end memory pipeline (drafting, rendering, budget control, and reading), training configurations with budget-aware objectives, evaluation protocols under extreme memory budgets, and baseline implementations.

### C.1. Technical Details

**End-to-end pipeline.** MemOCR follows a two-stage pipeline with an intermediate deterministic rendering step:

*Table 5.* Budget-to-resolution schedule used for MemOCR. The calculation is based on the 28×28 patch size of Qwen2.5-VL.

| Budget $\mathcal{B}$ | Target resolution (Max Number of Pixels) |
| --- | --- |
| 1024 | 802,816 |
| 256 | 200,704 |
| 64 | 50,176 |
| 16 | 12,544 |

- **Stage 1: Rich-text memory drafting.** We maintain a persistent rich-text memory $M_t^{\text{RT}}$ in Markdown. At each time step $t$, the agent takes the incoming chunk $C_t$ together with the current memory $M_{t-1}^{\text{RT}}$ and user query $Q$, and outputs an updated memory $M_t^{\text{RT}}$. Drafting process is *budget-agnostic*: the drafter writes a semantically complete memory while encoding priority via structure/formatting (*e.g.,* headings, bullets, and boldings).

- **Rendering (no LLM calls involved).** After processing all chunks, we convert the final rich-text memory $M_T^{\text{RT}}$ into a 2D memory image $V_T$ using a deterministic Markdown-to-image renderer with a fixed style sheet. The memory budget $\mathcal{B}$ is enforced *only at this step* by downsampling $V_T$ such that the image takes up at most $\mathcal{B}$ visual tokens.

- **Stage 2: Memory image reading.** The agent receives the budgeted memory image $V_T$ together with the question $Q$, and generates the final answer $A$. No raw history or original long context is provided at this stage, thus all task-relevant information must be recovered from $V_T$.

**Markdown rendering.** We implement a Markdown-to-image rendering module using FastAPI [1] and Playwright with Chromium backend [2]. Given an input Markdown string, the module (i) normalizes the text by stripping leading/trailing whitespace and surrounding backticks, (ii) converts Markdown to HTML using the Python `markdown` library [3], (iii) wraps the generated HTML in a fixed, inlined CSS template, and (iv) renders the HTML in a headless Chromium page and returns a screenshot image.

**Budget-to-resolution mapping.** Given a memory budget $\mathcal{B}$ (in *visual tokens*), we resize the rendered memory image to a target resolution such that the vision encoder produces $\leqslant \mathcal{B}$ tokens. We compute the visual token count using the backbone model's (*i.e.,* Qwen2.5-VL-7B-Instruct) patching size of 28×28=784 pixels per token. Table 5 provides the computed budget schedules used in our experiments.

**Prompt templates.** We use two prompt templates: (i) a **memory drafting** template that updates the Markdown memory given a problem, a new article chunk, and the previous memory; and (ii) a **memory reading** template that answers the problem based on the previous memory and wraps the answer in `\boxed{}`.

---

**Memory drafting prompt template.**

You are given a problem, an article, and a previous memory. You should draft the memory in markdown format with the crucial information in it that helps to answer the problem. In your markdown draft, you may use different headings to arrange the font sizes and styles of the information. *e.g.,* more important information should be emphasized and more visible (larger font size, bolder, etc.), in case the rendered image can be clearly read.
`<problem>` {PROBLEM} `</problem>` `<article>` {ARTICLE} `</article>` `<memory>` {MEMORY} `</memory>`
The draft memory, in markdown format:

---

**Memory reading prompt template.**

`<`{MEMORY IMAGE}`>`
You are presented with a problem and a previous memory. Please answer the problem based on the previous memory and put the answer in `\boxed{}`.
`<problem>`{PROBLEM}`</problem>`
Your answer:

---

[1] https://github.com/fastapi/fastapi
[2] https://github.com/microsoft/playwright
[3] https://pypi.org/project/Markdown/

*Table 6.* Budget-aware training task configuration.

| Task | Memory budget (Tokens) | Question source | Weight $w_k$ |
|------|------------------------|-----------------|--------------|
| $T_{\text{std}}$ | 512 | Original Question | 1.0 |
| $T_{\text{augM}}$ | 32 | Original Question | 0.7 |
| $T_{\text{augQ}}$ | 512 | Detail-oriented Question | 0.3 |

*Table 7.* Primary training hyperparameters for MemOCR.

| Category | Hyperparameter | Value |
|----------|----------------|-------|
| Algorithm | Chunk size $|C_t|$ | 5,000 |
| Algorithm | Group size $G$ | 16 |
| Algorithm | KL coefficient $\beta$ | $1 \times 10^{-3}$ |
| Algorithm | Clip ratio $\epsilon$ | 0.20 |
| Rollout | Top-$p$ | 0.999 |
| Rollout | Temperature | 1 |
| Rollout | Max rich-text memory tokens | 2048 |
| Rollout | Max final answer tokens | 2048 |
| Optimization | Global batch size | 64 |
| Optimization | Micro batch size | 16 |
| Optimization | Learning rate | $1 \times 10^{-6}$ |
| Optimization | Warmup Steps | 20 |

## C.2. Training Details

**Backbone models and parameter updates.** MemOCR is built on a vision-language model backbone Qwen2.5-VL-7B-Instruct. Unless otherwise stated, we train the model with full-parameter updates under BFloat16 precision under Fully Sharded Data Parallelism (FSDP) to scale to multi-GPU setups.

**Training data construction.** We train on long-context QA instances constructed from HotpotQA. For each training example, we assemble a long context by concatenating the supporting documents and sampled distractor documents, then pad/truncate to a target context length (*e.g.,* 30K tokens). We split the long context into a stream of chunks $\mathcal{C} = \{C_t\}_{t=1}^{T}$ using a fixed chunk size and stride, and update the persistent memory after each chunk.

**Budget-aware RL objectives.** We optimize the MemOCR layout/salience policy using a budget-aware RL objective with GRPO. We construct three training tasks: (i) standard QA with a moderate budget, (ii) QA under aggressively compressed memory images (low-budget robustness), and (iii) detail-oriented QA at high resolution (to ensure auxiliary details remain present but low-priority). We compute task-specific advantages for reader updates and use a weighted aggregation of advantages for drafting updates to learn a single layout policy that generalizes across budgets. Table 6 lists the task configurations and weights.

**Optimization and hyperparameters.** Table 7 summarizes the primary hyperparameters for GRPO algorithm, rollout generation, and optimization. Rollouts are generated with stochastic decoding (do_sample=*True*).

**Training hardware.** All three variants of MemOCR in Table 4 (full objectives, w/o $T_{\text{augM}}$ and w/o $T_{\text{augM}}, T_{\text{augQ}}$) are trained on 32 H800 NVIDIA GPUs for 7 days till convergence (about 1,000 steps, 21 $k$GPU-hours each).

## C.3. Evaluation Details

For evaluation, we construct contexts of approximately 10K/30K/100K tokens by concatenating the instance documents with sampled distractors, matching the training construction method. We fix the distractor sampling seed per split to ensure comparability across methods. Due to the time comsumption of long-context QA (as shown in Table 11), we follow Yu et al. to randomly downsample the four benchmarks to sizes of 128.

We use sub-word exact match (SEM) as accuracy and report the mean scores over three independent runs with different random seeds. A statistical significance analysis is conducted in Appendix D.1. Unless otherwise noted, we use stocastic decoding (temperature $= 0.7$, top-$p$=0.95) for both baselines and MemOCR.

### C.4. Baseline Setups

**Budget control for text baselines.**    For purely textual memory baselines, we enforce a token budget $\mathcal{B}$ by truncating the memory text to the first $\mathcal{B}$ tokens under a fixed tokenizer (Qwen2.5-7B-Instruct). We apply truncation at evaluation time only, using the last updated memory state (or the baseline's own memory update rule).

**Budget control for MemOCR.**    For MemOCR, we enforce the same budget $\mathcal{B}$ in visual tokens by resizing the rendered memory image such that the vision encoder produces $\leqslant \mathcal{B}$ visual tokens (Table 5). This ensures that comparisons reflect the same effective context capacity across text and vision modalities.

**Reproduction of baselines.**    We reproduce baseline memory systems according to their official releases:

- **MemAgent** (Yu et al., 2025): we use the authors' released code[4] and model weight[5] for both memory drafting and final answer generation. The chunk size $|C_t|$ is set to 5,000 following the authors' setup.

- **Mem$\alpha$** (Wang et al., 2025): we use the authors' released code[6] for reproduction. We conduct the memory drafting with the official model weight[7], and the final answer generation with Qwen2.5-7B-Instruct(Yang et al., 2024) to match the model size of other baselines. The chunk size $|C_t|$ is set to 1,000 following the authors' setup.

- **Mem0** (Chhikara et al., 2025): we run reproduction following the official documentation[8]. The memory drafting phase is finished with online black box APIs[9] and the answer generation with Qwen2.5-7B-Instruct. The chunk size $|C_t|$ is set to 5,000, consistent with our setup.

For the above three methods, we match (i) the answer generation model Qwen2.5-7B-Instrcut, (ii) the dataset split and long-context construction, and (iii) the decoding settings.

## D. Supplementary Experiments

### D.1. Statistical Verification

To ensure the reliability of our findings and rigorously validate the significance of the performance gains, we conduct a comprehensive statistical analysis. We perform independent experimental runs for both MemOCR and the textual baselines to capture variability. We report the mean scores (Mean) and standard deviations (Std) across all datasets and settings in Table 8. To formally assess the significiance of improvements, we perform an independent two-sample t-test on the averaged accuracy. The resulting $p$-values and performance gains ($\Delta$) are summarized in Table 9.

**Significance Analysis.**    The statistical data highlights two critical observations regarding the robustness and scalability of MemOCR. First, **MemOCR achieves consistent significant robustness under tight memory constraints.** As shown in Table 9, in low-memory scenarios ($\mathcal{B} \in \{16, 64\}$), the improvements are not only large in magnitude (*e.g.,* +30.6 at 10K/16 tokens) but also statistically significant, with $p$-values consistently far below the $0.05$ threshold (often $p \ll 0.01$). This confirms that the resilience of MemOCR against catastrophic forgetting is a systematic improvement. Second, **performance gaps diminish as the memory budget increases, and become marginal under the longest context.** At the maximum budget ($\mathcal{B} = 1024$), the gain becomes small at 100K context (only +0.9) and is not statistically significant ($p = 0.3419$), indicating saturation when both memory and context are ample. Meanwhile, MemOCR still shows modest but significant gains at shorter contexts (10K/30K), suggesting that the benefit is robust beyond extreme-budget regimes, while remaining most pronounced when the working context is severely constrained.

---

[4] https://github.com/BytedTsinghua-SIA/MemAgent
[5] https://huggingface.co/BytedTsinghua-SIA/RL-MemoryAgent-7B
[6] https://github.com/wangyu-ustc/Mem-alpha
[7] https://huggingface.co/YuWangX/Memalpha-4B
[8] https://github.com/mem0ai/mem0
[9] https://mem0.ai/

*Table 8.* Accuracies (Mean±Std) across all datasets, context lengths and memory budgets.

| Method | Memory Budget (Tokens) | HotpotQA 10K | 30K | 100K | 2Wiki 10K | 30K | 100K | NQ 10K | 30K | 100K | TriviaQA 10K | 30K | 100K | Average 10K | 30K | 100K |
|---|---|---|---|---|---|---|---|---|---|---|---|---|---|---|---|---|
| Mem0 | 1024 | 70.3±0.0 | 66.1±0.5 | 64.1±0.0 | 60.4±0.4 | 47.7±0.0 | 40.9±0.5 | 47.7±0.0 | 48.8±0.8 | 48.2±0.4 | 69.5±0.0 | 70.6±0.9 | 61.4±0.5 | 62.0±0.1 | 58.3±0.2 | 53.6±0.1 |
| | 256 | 71.1±0.0 | 65.6±0.0 | 63.8±0.5 | 57.8±0.0 | 44.8±0.5 | 43.5±0.5 | 46.9±0.0 | 51.1±0.5 | 49.7±0.9 | 69.5±0.0 | 70.6±0.5 | 62.8±0.5 | 61.3±0.0 | 58.0±0.2 | 55.0±0.3 |
| | 64 | 56.7±0.5 | 60.9±0.0 | 56.2±0.0 | 53.9±0.0 | 42.2±0.0 | 43.3±0.5 | 46.9±0.0 | 43.8±0.0 | 50.0±0.0 | 66.4±0.0 | 65.9±0.9 | 62.5±0.0 | 56.0±0.1 | 53.2±0.2 | 53.0±0.1 |
| | 16 | 37.0±0.5 | 43.3±0.5 | 34.9±0.5 | 28.4±0.5 | 31.5±0.5 | 27.6±0.5 | 41.4±0.0 | 34.1±0.9 | 43.0±0.0 | 65.3±0.5 | 62.8±0.9 | 58.9±0.5 | 43.0±0.1 | 42.9±0.3 | 41.1±0.2 |
| Mem-α | 1024 | 75.0±0.0 | 77.9±1.2 | 79.4±0.5 | 47.7±0.0 | 40.1±1.2 | 50.0±0.8 | 45.6±0.9 | 44.8±1.8 | 43.0±2.1 | 21.1±0.0 | 23.2±0.4 | 19.3±0.9 | 47.3±0.2 | 46.5±0.8 | 47.9±0.3 |
| | 256 | 57.8±1.6 | 57.3±0.9 | 57.0±1.4 | 38.3±0.8 | 32.8±0.8 | 41.7±0.5 | 47.9±0.9 | 33.6±0.8 | 38.0±2.5 | 17.2±0.8 | 24.2±1.4 | 18.3±0.5 | 40.3±0.2 | 37.0±0.9 | 38.7±0.9 |
| | 64 | 25.8±2.1 | 29.7±0.8 | 28.9±2.1 | 34.9±0.5 | 31.5±0.4 | 39.1±0.8 | 27.1±2.0 | 25.0±0.8 | 24.0±1.2 | 13.0±1.2 | 19.5±0.0 | 13.8±0.9 | 25.2±0.5 | 26.4±0.5 | 26.4±0.7 |
| | 16 | 24.0±1.2 | 30.2±1.2 | 26.3±0.9 | 35.4±0.9 | 32.3±0.9 | 40.1±0.5 | 23.7±0.9 | 26.3±0.5 | 24.2±1.3 | 11.7±0.8 | 19.0±1.6 | 14.3±0.9 | 23.7±0.6 | 27.0±0.5 | 26.2±0.1 |
| MemAgent | 1024 | 82.3±1.2 | 78.9±1.6 | 79.4±1.6 | 65.4±2.3 | 63.8±2.0 | 61.2±2.7 | 53.4±1.8 | 46.1±1.4 | 55.5±2.7 | 70.1±1.2 | 78.1±2.1 | 66.7±1.2 | 67.8±0.9 | 66.7±0.7 | 65.7±0.6 |
| | 256 | 82.3±1.2 | 76.8±2.4 | 76.6±1.4 | 66.1±3.3 | 62.5±1.4 | 58.1±2.5 | 51.3±2.0 | 44.3±1.6 | 53.6±2.5 | 69.5±2.1 | 77.6±1.2 | 67.2±2.1 | 67.3±1.4 | 65.3±1.5 | 63.9±0.5 |
| | 64 | 50.8±0.8 | 52.3±2.1 | 51.0±2.0 | 38.5±1.6 | 42.7±0.9 | 36.2±4.4 | 48.7±1.2 | 36.5±1.8 | 47.9±2.5 | 64.6±1.8 | 69.5±3.4 | 59.6±2.4 | 50.7±1.3 | 50.3±1.3 | 48.7±1.0 |
| | 16 | 24.0±1.2 | 26.6±0.8 | 23.2±0.5 | 28.4±0.5 | 26.3±1.6 | 16.7±0.5 | 24.5±1.6 | 20.1±1.2 | 27.3±2.1 | 49.7±1.2 | 56.3±1.4 | 41.7±1.8 | 31.6±0.5 | 32.3±0.4 | 27.2±0.3 |
| MemOCR | 1024 | 84.8±1.1 | 75.1±0.6 | 78.3±0.5 | 72.2±1.0 | 73.7±0.2 | 62.7±3.8 | 61.8±0.6 | 49.2±1.2 | 55.4±2.0 | 79.6±2.4 | 81.3±1.2 | 69.8±4.4 | 74.6±0.4 | 69.8±0.5 | 66.6±1.2 |
| | 256 | 82.2±1.8 | 75.4±0.5 | 75.7±1.3 | 71.2±0.3 | 72.8±1.8 | 65.5±3.6 | 57.3±2.2 | 48.8±1.3 | 58.3±1.8 | 79.7±1.8 | 80.9±1.8 | 68.9±3.1 | 72.6±0.7 | 69.5±0.9 | 67.1±0.7 |
| | 64 | 77.6±1.7 | 68.1±1.8 | 67.2±2.9 | 62.9±2.2 | 66.2±3.4 | 62.4±1.2 | 51.0±1.7 | 43.6±3.4 | 47.5±4.0 | 77.6±0.4 | 80.7±2.1 | 65.8±1.9 | 67.3±0.4 | 64.7±1.5 | 60.7±1.8 |
| | 16 | 67.2±4.7 | 57.9±6.0 | 52.4±1.0 | 57.9±1.0 | 56.0±2.3 | 45.7±3.5 | 42.8±1.9 | 35.9±7.9 | 45.9±4.3 | 80.8±2.6 | 72.2±5.7 | 67.2±3.8 | 62.2±0.7 | 55.5±1.7 | 52.8±0.4 |

*Table 9.* Statistical significance on average accuracy. **Bold** p-values indicate statistical significance ($p < 0.05$).

| Memory Budget(Tokens) | Ctx | MemOCR | MemAgent | Δ | p-value |
|---|---|---|---|---|---|
| | 10K | 74.6 ± 0.4 | 67.8 ± 0.9 | +6.8 | **0.0024** |
| 1024 | 30K | 69.8 ± 0.5 | 66.7 ± 0.7 | +3.1 | **0.0052** |
| | 100K | 66.6 ± 1.2 | 65.7 ± 0.6 | +0.9 | 0.3419 |
| | 10K | 72.6 ± 0.7 | 67.3 ± 1.4 | +5.3 | **0.0110** |
| 256 | 30K | 69.5 ± 0.9 | 65.3 ± 1.5 | +4.2 | **0.0217** |
| | 100K | 67.1 ± 0.7 | 63.9 ± 0.5 | +3.2 | **0.0032** |
| | 10K | 67.3 ± 0.4 | 50.7 ± 1.3 | +16.6 | **0.0008** |
| 64 | 30K | 64.7 ± 1.5 | 50.3 ± 1.3 | +14.4 | **0.0002** |
| | 100K | 60.7 ± 1.8 | 48.7 ± 1.0 | +12.0 | **0.0018** |
| | 10K | 62.2 ± 0.7 | 31.6 ± 0.5 | +30.6 | **<0.0001** |
| 16 | 30K | 55.5 ± 1.7 | 32.3 ± 0.4 | +23.2 | **0.0011** |
| | 100K | 52.8 ± 0.4 | 27.2 ± 0.3 | +25.6 | **<0.0001** |

## D.2. Computational Complexity Analysis

We analyze inference-time complexity in the long-horizon setting both theoretically and empirically. Our key finding is that MemOCR does not incur significant computation overhead compared to textual memory.

### D.2.1. THEORETICAL ANALYSIS

**Notation.** Let the full context contain $N$ tokens and be split into $T$ chunks $\mathcal{C} = \{C_t\}_{t=1}^{T}$, where each chunk contains $L \approx N/T$ tokens. Let $\mathcal{B}$ denote the memory budget used at the final answering step, as defined in §2. For simplicity, we ignore the question length which is typically small compared to memory. We assume a Transformer backbone with full self-attention, where a forward pass on length $x$ costs $\mathcal{O}(x^2)$ attention operations.

**Shared complexity: memory drafting in text domain.** At each step $t$, the agent consumes the current chunk and a bounded in-context memory:

$$M_t \sim \pi(\cdot \mid M_{t-1}, C_t), \quad t \in \{1, \ldots, T\}.$$

The memory shown to the updater can be estimated by

$$S_t = \mathcal{O}(|C_t| + |M_{t-1}|) = \mathcal{O}(L + \mathcal{B}),$$

where the memory budget $\mathcal{B}$ is the maximum number of in-context memory tokens. Therefore, the total drafting/update cost over $T$ chunks is

$$\text{Time}_{\text{draft}} = \sum_{t=1}^{T} \mathcal{O}(S_t^2) = \mathcal{O}\big(T(L + \mathcal{B})^2\big) = \mathcal{O}\Big(\frac{N}{L}(L + \mathcal{B})^2\Big). \tag{9}$$

Under our protocol where $L$ and $\mathcal{B}$ are fixed hyper-parameters, this stage scales *linearly* in the long context length $N$.

*Table 10.* Throughput and per-call latency of the renderer used in MemOCR (five runs on 50 samples each).

| Metric | Run 1 | Run 2 | Run 3 | Run 4 | Run 5 | Average |
|---|---|---|---|---|---|---|
| Throughput (Sample/s) | 68.1 | 70.6 | 67.9 | 70.8 | 63.7 | 68.2 |
| Latency (s) | 0.186 | 0.160 | 0.174 | 0.186 | 0.169 | 0.175 |

**Textual memory reading.** Textual-memory baselines answer by feeding a text memory of length $\mathcal{B}$ into the LLM:

$$A \sim \pi(\cdot \mid M_T, Q), \qquad |M_T| \leqslant \mathcal{B}.$$

Thus the answering cost is

$$\text{Time}_{\text{read, text}} = \mathcal{O}(\mathcal{B}^2). \tag{10}$$

**Visual memory reading.** MemOCR answers from a memory image $V_T$ whose vision encoder produces at most $\mathcal{B}$ visual tokens/patches:

$$A \sim \pi(\cdot \mid V_T, Q), \qquad \#\text{visual tokens}(V_T) \leqslant \mathcal{B}.$$

The cost in this stage consists of (1) the vision head processing the image (which is constant to $\mathcal{B}$ and $N$) and (2) attention over the $\mathcal{B}$ visual tokens in the language model, so the answering cost is

$$\text{Time}_{\text{read, visual}} = \mathcal{O}(\mathcal{B}^2). \tag{11}$$

**Overall theoretical complexity.** Combining Eq. (9)–(11) yields:

$$\text{TIME}_{\text{text}} = \underbrace{\mathcal{O}\big(T\,(L+\mathcal{B})^2\big)}_{\text{Time}_{\text{draft}}} + \underbrace{\mathcal{O}(\mathcal{B}^2)}_{\text{Time}_{\text{read, text}}} = \mathcal{O}\big(T\,(L+\mathcal{B})^2\big), \tag{12}$$

$$\text{TIME}_{\text{MemOCR}} = \underbrace{\mathcal{O}\big(T\,(L+\mathcal{B})^2\big)}_{\text{Time}_{\text{draft}}} + \underbrace{\mathcal{O}(\mathcal{B}^2)}_{\text{Time}_{\text{read, visual}}} = \mathcal{O}\big(T\,(L+\mathcal{B})^2\big). \tag{13}$$

Hence, MemOCR and textual memory have the same theoretical complexity scaling in $N$ (through $T \approx N/L$).

### D.2.2. EMPIRICAL ANALYSIS

**End-to-end runtime.** We report end-to-end runtime for MemOCR and a representative textual-memory baseline, MemAgent, under our long-context protocol. Table 11 summarizes the average per-instance runtime (seconds) across datasets under a 32-thread parallelism. The results exhibit the expected near-linear growth with context length (through $T$), and show that MemOCR remains comparable to (and even faster than on some cases) the text baseline at long contexts. The memory save comes majorly from the shorter memory states – we find our model sometimes generate less tokens in memory states than the textual memory baseline, which exceed the visual encoding overhead.

**Visual-only overhead: memory image rendering.** MemOCR additionally renders $M_T^{\text{RT}}$ into a memory image before memory reading. This step is deterministic and does not invoke any LLM, and its runtime is linear in the output canvas size. We empirically study its computational overhead by feeding 50 rich-text memory samples into the renderer for 5 times, and the results are in Table 10. The results indicate the image rendering process is very light-weighted, consuming only 1 second per 68 samples and a 0.175 extra latency.

### D.3. Additional Ablation Studies

**Motivation.** Table 4 shows that removing RL from our 7B setting causes substantial degradation, especially under strict memory budgets. A natural question is whether simply scaling the backbone LLM can close the gap without our budget-aware RL. To answer this, we further report results using Qwen2.5-VL-32B-Instruct and Qwen2.5-VL-72B-Instruct under the same memory-budgeted evaluation protocol. We follow the same evaluation setup as in §4.5.

**Key observations.** **(1) 7B+RL beats scaling.** Despite being trained on a 7B backbone, MemOCR matches or exceeds the scaled non-RL backbones (32B/72B) in most long-context settings, indicating that budget-aware RL is more effective than

*Table 11.* Average runtime (second/sample) of MemOCR and MemAgent across different datasets and memory budgets.

| Method | Memory Budget (Tokens) | HotpotQA | | | 2Wiki | | | NQ | | | TriviaQA | | | Average | | |
|---|---|---|---|---|---|---|---|---|---|---|---|---|---|---|---|---|
| | | 10K | 30K | 100K | 10K | 30K | 100K | 10K | 30K | 100K | 10K | 30K | 100K | 10K | 30K | 100K |
| MemOCR | 1024 | 19.5 | 40.8 | 173.7 | 11.3 | 44.0 | 187.5 | 13.5 | 47.8 | 240.1 | 15.7 | 58.5 | 241.2 | 15.0 | 47.8 | 210.6 |
| | 256 | 12.7 | 39.7 | 173.9 | 10.1 | 45.4 | 180.8 | 12.3 | 47.8 | 231.4 | 15.3 | 57.5 | 229.1 | 12.6 | 47.6 | 203.8 |
| | 64 | 11.6 | 39.9 | 167.8 | 9.6 | 43.7 | 182.6 | 9.8 | 48.3 | 235.7 | 16.3 | 51.0 | 229.9 | 11.8 | 45.7 | 204.0 |
| | 16 | 9.9 | 37.7 | 174.9 | 10.0 | 47.5 | 186.1 | 9.1 | 47.0 | 207.8 | 8.5 | 59.8 | 208.3 | 9.4 | 48.0 | 194.3 |
| MemAgent | 1024 | 8.9 | 41.6 | 208.2 | 6.6 | 35.9 | 306.6 | 9.1 | 47.3 | 230.2 | 9.4 | 49.5 | 232.0 | 8.5 | 43.6 | 244.3 |
| | 256 | 8.8 | 42.0 | 207.6 | 6.7 | 35.5 | 305.8 | 9.2 | 47.7 | 228.9 | 9.4 | 49.9 | 232.7 | 8.5 | 43.8 | 243.8 |
| | 64 | 8.9 | 41.6 | 205.7 | 6.7 | 36.0 | 300.6 | 9.2 | 47.5 | 226.8 | 9.4 | 49.8 | 231.7 | 8.5 | 43.7 | 241.2 |
| | 16 | 8.7 | 42.0 | 206.5 | 6.6 | 35.9 | 301.3 | 9.4 | 47.3 | 229.4 | 9.3 | 49.4 | 231.1 | 8.5 | 43.6 | 242.1 |

*Table 12.* Accuracies (%) under larger backbones (Qwen2.5-VL-32/72B-Instruct) without RL.

| Method | Memory Budget (Tokens) | HotpotQA | | | 2Wiki | | | NQ | | | TriviaQA | | | Average | | |
|---|---|---|---|---|---|---|---|---|---|---|---|---|---|---|---|---|
| | | 10K | 30K | 100K | 10K | 30K | 100K | 10K | 30K | 100K | 10K | 30K | 100K | 10K | 30K | 100K |
| **MemOCR (Ours)** | 1024 | **84.8** | 75.1 | **78.3** | **72.2** | **73.7** | 62.7 | **61.8** | 49.2 | 55.4 | 79.6 | 81.3 | 69.8 | **74.6** | **69.8** | 66.6 |
| | 256 | 82.2 | **75.4** | 75.7 | 71.2 | 72.8 | **65.5** | 57.3 | 48.8 | **58.3** | 79.7 | 80.9 | 68.9 | 72.6 (-2.7%) | 69.5 (-0.5%) | **67.1** (+0.8%) |
| | 64 | 77.6 | 68.1 | 67.2 | 62.9 | 66.2 | 62.4 | 51.0 | 43.6 | 47.5 | 77.6 | 80.7 | 65.8 | 67.3 (-9.8%) | 64.7 (-7.4%) | 60.7 (-8.8%) |
| | 16 | 67.2 | 57.9 | 52.4 | 57.9 | 56.0 | 45.7 | 42.8 | 35.9 | 45.9 | **80.8** | 72.2 | 67.2 | 62.2 (-16.6%) | 55.5 (-20.5%) | 52.8 (-20.7%) |
| Qwen-VL-7B | 1024 | 67.2 | 54.7 | 41.4 | 51.6 | 45.3 | 38.3 | 45.3 | 40.6 | 43.8 | 62.5 | 68.0 | 47.7 | 56.6 | 52.1 | 42.8 |
| | 256 | 62.5 | 51.6 | 46.9 | 46.9 | 46.9 | 31.3 | 46.9 | 36.7 | 42.2 | 71.1 | 70.3 | 58.6 | 56.8 (+0.3%) | 51.4 (-1.5%) | 44.7 (+4.6%) |
| | 64 | 62.5 | 50.8 | 42.2 | 47.7 | 43.0 | 32.8 | 46.9 | 36.7 | 39.8 | 66.4 | 67.2 | 53.1 | 55.9 (-1.4%) | 49.4 (-5.2%) | 42.0 (-1.8%) |
| | 16 | 33.6 | 30.5 | 28.9 | 34.4 | 25.0 | 28.1 | 27.3 | 25.0 | 28.1 | 54.7 | 60.2 | 46.9 | 37.5 (-33.8%) | 35.2 (-32.6%) | 33.0 (-22.8%) |
| Qwen-VL-32B | 1024 | 64.8 | 59.4 | 53.1 | 60.2 | 54.7 | 33.6 | 50.8 | 48.4 | 50.0 | 77.3 | 75.0 | 70.3 | 63.3 | 59.4 | 51.8 |
| | 256 | 71.9 | 60.2 | 56.3 | 60.2 | 50.8 | 40.6 | 50.0 | 42.2 | 48.4 | 74.2 | 76.6 | 69.5 | 64.1 (+1.2%) | 57.4 (-3.3%) | 53.7 (+3.8%) |
| | 64 | 58.6 | 57.0 | 44.5 | 51.6 | 43.0 | 30.5 | 46.1 | 45.3 | 43.8 | 73.4 | 71.9 | 66.4 | 57.4 (-9.3%) | 54.3 (-8.6%) | 46.3 (-10.6%) |
| | 16 | 31.3 | 31.3 | 34.4 | 32.0 | 32.8 | 21.9 | 37.5 | 35.2 | 35.9 | 63.3 | 66.4 | 59.4 | 41.0 (-35.2%) | 41.4 (-30.3%) | 37.9 (-26.8%) |
| Qwen-VL-72B | 1024 | 78.1 | 60.9 | 60.2 | 68.8 | 53.9 | 37.5 | 57.8 | 51.6 | 46.1 | 78.1 | **82.8** | **71.1** | 70.7 | 62.3 | 53.7 |
| | 256 | 75.8 | 65.6 | 57.8 | 63.3 | 57.8 | 46.9 | 60.2 | **56.3** | 50.0 | 79.7 | 82.0 | **71.1** | 69.7 (-1.4%) | 65.4 (+5.0%) | 56.4 (+5.1%) |
| | 64 | 66.4 | 60.2 | 50.8 | 54.7 | 46.1 | 34.4 | 54.7 | 46.9 | 46.1 | 77.3 | 77.3 | 68.8 | 63.3 (-10.5%) | 57.6 (-7.5%) | 50.0 (-6.9%) |
| | 16 | 35.9 | 42.2 | 43.8 | 24.2 | 25.8 | 23.4 | 47.7 | 39.8 | 37.5 | 69.5 | 68.8 | 63.3 | 44.3 (-37.3%) | 44.1 (-29.2%) | 42.0 (-21.8%) |

naïve model scaling. **(2) Lower decay under compression.** When shrinking the memory budget, MemOCR shows much smaller relative drops than all non-RL baselines, and remains reliable even at the extreme 16-token budget where larger backbones degrade sharply.

textbf(3) Strength on long-context multi-hop. The advantage is most pronounced at 100K context on multi-hop benchmarks (HotpotQA, 2Wiki), suggesting the gain comes from robust memory usage rather than short-context capacity.

### D.4. Cross-Model Transfer

We test whether MemOCR's visual memory transfers across VLMs by having GPT-4o read memory images generated by MemOCR. Table 13 reports accuracy on HotpotQA (Yang et al., 2018) at three context lengths. GPT-4o equipped with MemOCR's memory achieves 87.7 at 10K, surpassing even Qwen2.5-VL (Bai et al., 2025) as the native reader (84.8). The layout-aware variant consistently outperforms the w/o Layout variant, while Mem-$\alpha$ (Wang et al., 2025) memory yields substantially lower scores. These results confirm that both the typographic hierarchy and spatial layout generalize to a reader that was never involved in memory construction.

*Table 13.* Cross-model transfer on HotpotQA. GPT-4o serves as the reader for all memory sources.

| Memory Source | 10K | 30K | 100K |
|---|---|---|---|
| MemOCR's Memory | 87.7 | 83.5 | 82.9 |
| MemOCR w/o Layout | 84.4 | 81.5 | 80.1 |
| Mem-$\alpha$'s Memory | 65.6 | 57.0 | 63.3 |
| w/o Memory | 53.9 | 49.4 | 47.2 |

### D.5. Aspect Ratio Robustness

We test MemOCR under different rendering aspect ratios to verify that the layout policy is not brittle to image shape. Table 14 reports accuracy averaged over all four benchmarks. All three configurations yield comparable performance with less than 2 points of difference across every setting, suggesting that the VLM reader relies primarily on typographic cues (font weight, size) rather than absolute spatial coordinates.

Table 14. Aspect ratio robustness. Accuracy averaged over HotpotQA, 2WikiMultiHopQA, NQ, and TriviaQA.

| Aspect Ratio | 10K | 30K | 100K |
|---|---|---|---|
| Fixed width (original) | 74.6 | 69.8 | 66.6 |
| Square | 74.6 | 70.9 | 67.0 |
| 1:10 stretching | 72.6 | 69.2 | 68.3 |

### D.6. Alignment Tax

A natural concern is whether RL training degrades general language understanding. We evaluate MemOCR on MMLU and observe only a 1.4-point drop relative to the base Qwen2.5-VL-7B-Instruct (Bai et al., 2025) model (Table 15). This minimal degradation indicates that the RL training stage focuses narrowly on memory-related capabilities, leaving broader language understanding largely intact.

Table 15. Alignment tax measured by MMLU accuracy.

| Model | MMLU |
|---|---|
| Qwen2.5-VL-7B-Instruct | 72.5 |
| MemOCR | 71.1 |

### D.7. RL Training Data Ablation

We investigate the effect of training data composition by replacing HotpotQA (Yang et al., 2018) (multi-hop) with NQ (Kwiatkowski et al., 2019) (single-hop) for 1k gradient steps. As shown in Table 16, training on NQ improves NQ performance substantially (66.1 vs 45.3 at $\mathcal{B}$=1024) but degrades HotpotQA below the no-RL baseline (53.9 vs 67.2). In contrast, training on HotpotQA yields strong gains on both benchmarks (+17.6 on HotpotQA, +16.5 on NQ at full budget). This suggests that multi-hop data teaches a richer drafting policy that transfers to simpler tasks, whereas single-hop training overfits to a narrow distribution.

Table 16. RL training data ablation (10K context).

| Training Data | Budget | HotpotQA | NQ |
|---|---|---|---|
| None (no RL) | 1024 | 67.2 | 45.3 |
| None (no RL) | 16 | 33.6 | 27.3 |
| NQ | 1024 | 53.9 | 66.1 |
| NQ | 16 | 32.8 | 45.3 |
| HotpotQA (ours) | 1024 | 84.8 | 61.8 |
| HotpotQA (ours) | 16 | 67.2 | 42.8 |

### D.8. Training Convergence

We compare training reward curves of MemOCR and MemAgent (Yu et al., 2025) over 1000 GRPO steps, both optimizing exact-match reward on HotpotQA (Yang et al., 2018) with identical hyperparameters (Table 17). MemAgent converges slightly faster in the first 400 steps, likely because its text-based memory format aligns more closely with the pretrained distribution. However, MemOCR overtakes after approximately 600 steps and maintains a growing advantage through convergence, suggesting that the visual memory paradigm offers a higher performance ceiling once the policy adapts to the rendered format.

## E. Bad Case Analysis

While MemOCR demonstrates strong performance in identifying and emphasizing crucial information via layout generation, it also fail on certain data points. In Figure 8, we analyze two representative failure modes observed during our experiments.

*Table 17.* Training reward (exact-match on HotpotQA) over RL steps.

| Steps | MemOCR | MemAgent |
|-------|--------|----------|
| 200   | 0.669  | 0.687    |
| 400   | 0.698  | 0.706    |
| 600   | 0.734  | 0.729    |
| 800   | 0.761  | 0.752    |
| 1000  | 0.792  | 0.782    |

**Failure Mode A: Loss of fine-grained details in comparative reasoning.** The first type of failure occurs when the question requires comparing detailed attributes of two entities, but the agent layout that prioritizes the entity names over the descriptions. As shown in the top panel of Figure 8, for the question *"Who has a wider scope of profession...?"*, the agent correctly identifies "Hrag Vartanian" and "Hovsep Pushman" as key entities and renders them as H1 headers. However, the specific details required for comparison are rendered as standard body text. Under the constraint of a low token budget (resulting in a low-resolution downsampled image), the large headers remain legible, but the smaller body text collapses into unreadable pixel noise, ultimately leading to an incorrect answer.

**Failure Mode B: Information loss due to memory capacity overflow.** The second failure mode arises when the rich-text memory is excessively long. This sometimes happens due to some generation issue where the model repeats the same words for multiple times, forcing the rendering engine to compress the font size below the readability threshold of the visual encoder. In the bottom panel of Figure 8, the agent attempts to encode a lengthy history of "Adidas Yeezy" (over 2000 characters) into a single memory image. The font size is drastically reduced after downsampling to a fixed number of pixels. Consequently, the crucial evidence, such as the specific release date "February 14, 2015", becomes indistinguishable. Unlike Failure Mode A, where headers preserved some partial information, this case may result in total information loss.

## Failure Mode A: Loss of Fine-grained Details in Comparative Reasoning

**Both Entity Visible, but no Details**

**Question:** *Who has a wider scope of profession, Hrag Vartanian or Hovsep Pushman?*

Ground Truth:
**Hrag Vartanian**

Downsampled Memory Image

The Answer is
**Hovsep Pushman**.

## Failure Mode B: Memory Capacity Overflow

**Key Entity (2015) Unreadable**

**Question:** *When did the first pair of yeezys come out?*

Ground Truth:
**February 14, 2015**

The Answer is
**February 2012**.

Downsampled Memory Image

*Figure 8.* **Failure Mode Analysis under Resource Constraints (16-token budget).** (Top) In comparative reasoning (*i.e.,* to choose among two candidates), while the layout successfully highlights entity headers, the body text containing crucial attributes is compressed into unreadable noise during downsampling. (Bottom) When the rich-text memory length exceeds the visual canvas capacity, the forced font scaling drops below the visual encoder's resolution threshold, resulting in information loss.

