# OpenReview forum: "MemOCR: Layout-Aware Visual Memory for Efficient Long-Horizon Reasoning"
_ICML.cc/2026/Conference — ICML 2026 regular_

### Official Review · Reviewer_ATJ4 · 2026-02-16

**Soundness:** 1
**Presentation:** 1
**Significance:** 2
**Originality:** 2
**Overall Recommendation:** 2
**Confidence:** 5

**Summary:**

This paper proposes MemOCR, a layout-aware visual memory framework for long-horizon LLM agents. Unlike retrieval-augmented generation (RAG) systems, MemOCR does not rely on external retrieval but instead maintains a compressed internal memory representation. Rather than storing memory as uniform-cost text, it drafts structured rich text with explicit visual salience (e.g., headings, highlights), renders it into an image, and performs inference under a controllable visual-token budget. With budget-aware RL training, the model learns to allocate visual space non-uniformly so that crucial evidence remains readable under compression. Experiments on long-context QA benchmarks show strong improvements over text-based memory baselines, particularly under extreme budget constraints, and ablations suggest that layout control plays a key role in robustness.

**Compliance With Llm Reviewing Policy:**

Affirmed.

**Final Justification:**

For the following reasons, I strongly recommend rejection and encourage the authors to clearly articulate their contributions and substantially revise the paper.

My first concern is that methods originally designed and optimized under their own settings are being applied under the proposed evaluation protocol of this paper without appropriate adaptation, which leads to a seriously unfair comparison. In the original papers, the major baselines rely on vector databases with top-k retrieval, or are optimized for a specific memory budget (e.g., 1024 tokens). However, in this work, these methods are evaluated by either removing retrieval entirely or applying them under different token budgets without proper re-optimization. In particular, to meet the token budget, the baselines are simply truncated, which fails to reflect their intended usage, resulting in a seriously unfair comparison.

Next, although the proposed method clearly builds upon Visual-Text Compression (VTC) approaches and the authors are aware of prior VTC work, (1) there is very limited discussion of VTC in the Introduction and throughout the paper, which makes it difficult to clearly identify the paper’s contributions and may even lead to the impression that VTC itself is part of the contribution, and (2) there is no experimental comparison with prior VTC methods in the original submission. I consider this to be a serious issue, as it makes it difficult to determine what the paper actually contributes.

While the rebuttal provides some meaningful additions in response to the reviewer’s concerns, addressing these issues would require substantial revisions to the experimental section, and the Introduction would also need to be significantly rewritten to properly position the work with respect to prior VTC research. Therefore, I remain highly skeptical about the suitability of this paper for acceptance.

**Key Questions For Authors:**

- **(Q1)** The memory budgets (8/16/64/256/1024) are extremely aggressive and may not reflect practical settings, especially compared to typical RAG pipelines that can store long histories externally. Can the proposed approach be integrated into a RAG-style system?

- **(Q2)** It is also unclear whether a compressed visual memory can reliably store more useful information than a well-optimized textual summary under the same effective budget. A direct comparison against strong text summarization baselines (rather than truncation-based variants) would help clarify this point.

**Limitations:**

Yes, the limitations are discussed in Appendix B.

**Strengths And Weaknesses:**

**Strengths**
- **(S1)** The method compresses memory by converting text into a visual format and then downsampling it to fit a given budget. The idea of preserving critical information through visual salience (e.g., larger fonts, bold text) is interesting.
- **(S2)** Because memory is stored as an image, the system can adapt to different memory budgets by simply changing the downsampling level.


**Weaknesses**
- **(W1)** The evaluation protocol raises concerns about fairness. Mem0 and Mem-α are RAG-style methods. They build a vector database and answer via retrieval. However, the paper compares them by not using retrieval and instead truncating their memory to the budget. This is not how these methods are intended to work. Similarly, MemAgent is optimized around a specific memory budget (e.g., 1024), but the paper reduces budgets by naive truncation. Text-based methods are not optimized for “truncate-to-budget,” so Table 1 likely overstates MemOCR’s advantage. This evaluation setup may make the proposed method look stronger than it actually is under extreme compression.

- **(W2)** For a fair comparison, Mem0 and Mem-α should be evaluated in their intended retrieval-based setting, where the memory budget constrains the amount of retrieved evidence (e.g., top-k results) rather than truncating the entire memory state. Shrinking the vector database itself may not reflect realistic usage, and forcibly truncating retrieved results solely to match a token budget could lead to an evaluation that does not align with how retrieval-based systems are designed to operate.

- **(W3)** MemAgent should not be adapted to different budgets by truncation either. It should be re-optimized (or at least properly tuned) for each budget.

- **(W4)** More experiments on long-term conversation benchmarks (e.g., LOCOMO) would strengthen the paper, as such settings may more clearly reveal the limitations of the proposed approach compared to RAG-based methods.

- **(W5)** It would be more compelling if the proposed method were evaluated as a component within a RAG-style pipeline, rather than as a replacement for retrieval-based memory. This would clarify whether visual compression provides complementary benefits beyond standard retrieval and summarization mechanisms.

- **(W6)** In Table 1, the baselines use Qwen2.5-Instruct, while MemOCR uses Qwen2.5-VL. The paper should analyze whether backbone differences contribute to the gains.

- **(W7)** More qualitative analysis beyond Fig. 8 would be helpful, particularly additional examples illustrating how the visual memory is structured and which information is emphasized versus compressed.

- **(W8)** While downsampling enables budget adaptation, heavy downsampling may make text unreadable (information collapse). In such cases, token-based text memory could be more reliable.

- **(W9)** Lines 767–769 mention downsampling to 128 samples following Yu et al., but the paper should explain this choice more clearly.

- **(Minor-W10)** In Table 5, “Value (TBD)” looks like a placeholder or typo.

---

> ### Author Rebuttal · Authors · 2026-03-31
>
> We thank for the detailed review and address each concern below.
>
> > **W1.**
>
> We fully agree that fairness limitations in the MemAgent experiments (w.r.t. **Q2**), though we politely insist that Mem-α/Mem0 comparisons are relatively fair (w.r.t. **W2**).
>
> > **W2.**
>
> In our original setting, **"memory budget" constrains tokens consumed in context, not the external vector database**. At the reviewer's advice, we conduct **additional experiments** evaluating Mem0 and Mem-α in their intended retrieval-based mode, where the budget constrains $k$ in top-k retrieval rather than truncating after retrieval.
>
> **Table R1.** Comparison with RAG methods (10K context) on HotpotQA.
>
> |Budget|Mem0(ret)|Mem0(trunc)|Mem-α(ret)|Mem-α(trunc)|MemOCR|
> |---|---|---|---|---|---|
> |1024|66.1|70.3|49.2|75.0|**84.8**|
> |256|62.5|71.1|29.7|57.8|**82.2**|
> |64|56.5|56.7|7.0|25.8|**77.6**|
> |16|54.4|37.0|5.5|24.0|**67.2**|
>
> Key observations:
> 1. The **trunc**ation mode is generally more beneficial for the two RAG baselines.
> 2. Mem-α performs particularly poorly in **ret**rieval mode because its memory pieces are large and can be entirely truncated out ($k=0$) under low budgets.
>
> > **W3**
>
> Please see our response to **Q2**.
>
> > **W4.**
>
> We have conducted zero-shot evaluations on LOCOMO [1]:
>
> **Table R2.** Zero-shot evaluation on LOCOMO.
>
> |Budget|Mem-α|MemOCR|
> |---|---|---|
> |1024|23.83|**33.14**|
> |256|23.05|**31.71**|
> |64|8.59|**17.52**|
> |16|3.12|**15.63**|
>
> MemOCR outperforms Mem-α by ~9 points at full budget and maintains strong performance under extreme compression, confirming generalization to conversational memory settings.
>
> > **W5**
>
> Please see our response to **Q1**.
>
> > **W6.**
>
> We **conduct new experiments** evaluating Mem0 and Mem-α with the same Qwen2.5-VL backbone as MemOCR:
>
> **Table R3.** Qwen2.5-VL-7B on HotpotQA, 1024 budget.
>
> |Method(w/ Qwen2.5-VL)|10K|30K|100K|
> |---|---|---|---|
> |Mem0 w/ VL-7B|66.1|46.4|52.9|
> |Mem-α w/ VL-7B|49.2|39.8|52.3|
> |MemOCR|**84.8**|**75.1**|**78.3**|
>
> Mem0 and Mem-α perform worse with VL than with Instruct backbone in the original Table 1, confirming that gains are not attributable to backbone differences.
>
> > **W7.**
>
> We are preparing additional qualitative examples in the revised appendix, including side-by-side memory images at different budgets and examples showing how query types affect layout.
>
> > **W8.**
>
> We fully agree with this limitation. Mitigating information collapse under heavy downsampling is precisely what our layout policy aims to address.
> Unlike text summarization which suits text-based memory, MemOCR allocates visual saliency to high-priority information of visual memory to mitigate the information collapse.
>
> > **W9.**
>
> We apologize for the unclear description. In the recurrent paradigm, the agent needs 20+ turns of interaction for 100K context (~200s/sample on 8 GPUs in Table 9). We follow Yu et al. [3] to downsample due to heavy inference cost. Statistical verification has been conducted to guarantee confidence on the subset.
>
> > **W10 (Minor).**
>
> Thank you for catching this. We will correct this in the revision.
>
> > **Q1.** The memory budgets are extremely aggressive ...
>
> We fully agree that extremely low budgets (e.g., 16 tokens) may not reflect RAG needs.
>
> In our experiments, we report scores under extreme memory budgets because we surprisingly find out MemOCR achieves only ~20% accuracy drop under 128× compression.
> This suggests its potential that by scaling up the budget, we may encode tens or hundreds of retrieved documents into 1024 visual tokens within a RAG pipeline.
>
> > Can the proposed approach be integrated into a RAG-style system?
>
> Yes. MemOCR is an independent memory compression model that can serve as a solution for the over-length retrieval problem in RAG systems [2].
> We are conducting experiments to evaluate MemOCR's boosts in RAG pipelines, and the results will be updated once we have them.
>
> > **Q2.** A direct comparison against strong text summarization baselines would help clarify this point.
>
> In our original setup, we allows 1024 tokens for intermediate memory states $m_t$ ($1\leq t < T$), and truncate the last memory state to $B$ tokens.
>
> We have conducted **additional experiments** evaluating MemAgent with text summarization as memory compression.
> Here we use a Qwen2.5-7B-instruct model to summarize the last memory state, instead of truncating.
>
> **Table R4.** Comparison with text-summarization. Scores are avg performance on 4 benchmarks under 10K context.
>
> |Budget|MemOCR|MemAgent|
> |---|---|---|
> |1024|**74.6**|66.8|
> |256|**72.6**|67.2|
> |64|**67.3**|62.0|
> |16|**62.2**|58.5|
>
> Key observations:
> 1. Summarization is indeed a fairer implementation for text-based baselines.
> 2. The same trends as Table 1 hold: MemOCR outperforms baseline method under all budgets.
>
> [1] Evaluating Very Long-Term Conversational Memory of LLM Agents
>
> [2] Scaling Long-Horizon Agent via Context Folding
>
> [3] MemAgent: Reshaping Long-Context LLM with Multi-Conv RL-based Memory Agent

---

> > ### Author Rebuttal · Reviewer_ATJ4 · 2026-04-03
> >
> > I believe that the rebuttal does not sufficiently address the core weaknesses I raised. Since substantial revisions to the experimental section are required, it is risky to assume that the paper can resolve these issues in its current form.
> > - As the authors themselves acknowledge, key parts of the experimental setup are fundamentally unfair. Although some additional experiments were presented following my suggestions, I believe that many aspects of the evaluation protocol would need to be replaced rather than incrementally improved. Therefore, there are too many issues that require revision to justify acceptance based solely on the rebuttal.
> > - Despite the fact that Visual-Text Compression approaches such as DeepSeek-OCR and Glyph have already been proposed in prior work, the original paper does not include comparisons against these methods. While such comparisons were partially introduced in the rebuttal, this still implies that a major revision of the experimental section is necessary rather than a minor addition.
> > - While the method is positioned as a novel visual memory paradigm, it fundamentally builds upon existing Visual-Text Compression (VTC) approaches (e.g., DeepSeek-OCR, Glyph). However, the paper does not clearly disentangle what is genuinely new beyond prior VTC methods. In particular, the Introduction contains very limited discussion of prior VTC approaches. This makes it difficult to clearly assess the paper’s contribution relative to existing work, and addressing this issue would require a substantial revision of the Introduction.
> >
> > For these reasons, I am inclined to strongly recommend rejection of this paper. I will lower my score, and encourage the authors to clearly distinguish their contributions from prior VTC methods and to substantially revise the experimental section before submitting to a future venue.

---

> > > ### Author Response · Authors · 2026-04-04
> > >
> > > We acknowledge the Reviewer's follow-up response and provide a final clarification on the core disagreement.
> > >
> > > **On the experimental comparison protocol.** The research question of this paper is to compare the efficiency of different memory compression schemes under a given context token budget. Comparing methods under a unified token budget is the natural experimental design that directly serves this research question — in fact, this is also one of the key metrics reported in the Mem-α paper [1].
> > >
> > > The Reviewer prefers evaluating Mem0 and Mem-α in a RAG-style top-k retrieval mode. We understand this as another evaluation dimension that is inherently biased toward the retrieval paradigm, rather than the only correct comparison protocol. Moreover, our experiments show that this retrieval mode is actually *less favorable* to the baselines.
> > >
> > > **Throughout the rebuttal**, we have conducted targeted experiments addressing the vast majority of the Reviewer's concerns: retrieval-based evaluation (Table R1), text summarization replacing truncation (Table R4), unified Qwen2.5-VL backbone (Table R3), and 4× token budget compensation. Every new experiment yields conclusions fully consistent with the original paper.
> > >
> > > We acknowledge that some of these additional experiments help strengthen the evaluation. However, labeling the original experiments as "fundamentally unfair" solely because they do not follow one particular evaluation protocol is irresponsible and unprofessional. This does not constitute a valid reason for Strong Reject. By ICML's overall recommendation definition, Strong Reject applies to "a paper with well-known results, unaddressed ethical considerations, or a poorly written paper where it is impossible to tell what the nature of its contribution is." This paper (1) presents new experimental results, (2) has no unaddressed ethical considerations, and (3) received agreeing recognition for its presentation quality (four `good` Presentation scores from all four reviewers).
> > >
> > > We believe this review is unreasonable and seriously deviates from the ICML Reviewing Principles. We urge Reviewer ATJ4 to reconsider the contributions of this paper.
> > >
> > > [1] Mem-$\alpha$: Learning memory construction via reinforcement learning

---

### Official Review · Reviewer_4rjh · 2026-03-06

**Soundness:** 3
**Presentation:** 3
**Significance:** 3
**Originality:** 2
**Overall Recommendation:** 4
**Confidence:** 4

**Summary:**

The paper introduces MemOCR, a novel multimodal memory agent designed to optimize long-horizon reasoning under strictly constrained context windows. Addressing the inherent limitation of text-based memory, where token cost scales linearly regardless of information importance, MemOCR shifts to a visual memory paradigm. The agent drafts a rich-text (Markdown) summary of its history, strategically using typographic formatting to assign visual prominence to crucial evidence. This text is then rendered into a 2D image, allowing the agent to dynamically control the context budget via image resolution adjustments. Trained using GRPO across tasks with varying compression levels, MemOCR learns to achieve adaptive information density, significantly outperforming text-based baselines under tight memory constraints.

**Compliance With Llm Reviewing Policy:**

Affirmed.

**Final Justification:**

I have read the authors’ response and the additional experiment. I have raised my score to weak accept.

**Key Questions For Authors:**

- The layout policy is heavily optimized for long-context QA tasks. It is unclear how well this specific salience allocation strategy transfers to more dynamic agentic workloads, such as continuous planning or tool-use logs, where importance is highly contextual and temporal.
- The core conceptual leap is somewhat marginal. The methodology reads heavily like a straightforward pipeline integration of DeepSeek-OCR and established text-based memory summarization paradigms.
- While the paper targets long-horizon reasoning, the evaluation heavily relies on concatenated multi-hop and single-hop QA datasets. These traditional QA tasks often fail to capture the true complexity of long-context understanding, information synthesis, and temporal reasoning over extended sequences. The paper lacks evaluation on long-context benchmarks such as LOCOMO [1], which are the standard testing grounds for evaluating memory limits and retrieval capabilities in modern long-horizon tasks.
-The paper lacks a comparative analysis against the most recent baselines. Without comparing against the current SOTA [2,3,4], it is difficult to accurately gauge the absolute performance ceiling and the relative superiority of the MemOCR architecture.
- Overall, I think the paper is not bad. If the author can fix these problems, I’m open to raise my score.


[1] Maharana, Adyasha, et al. "Evaluating very long-term conversational memory of llm agents." Proceedings of the 62nd Annual Meeting of the Association for Computational Linguistics (Volume 1: Long Papers). 2024.

[2] Fang, Jizhan, et al. "Lightmem: Lightweight and efficient memory-augmented generation." arXiv preprint arXiv:2510.18866 (2025).

[3] Zhao, Xiaochen, et al. "HyMem: Hybrid Memory Architecture with Dynamic Retrieval Scheduling." arXiv preprint arXiv:2602.13933 (2026).

[4] Liu, Jiaqi, et al. "SimpleMem: Efficient Lifelong Memory for LLM Agents." arXiv preprint arXiv:2601.02553 (2026).

**Limitations:**

Yes

**Strengths And Weaknesses:**

- Transitioning from a 1D token stream to a 2D visual canvas is a creative and highly effective solution to the uniform information density problem. Decoupling semantic content from context cost allows for highly efficient budget allocation.
- The empirical results are compelling, particularly the model's graceful degradation under extreme token constraints.
- The budget-aware RL objectives cleverly prevent the agent from collapsing into a uniform layout policy, ensuring that both critical visibility and auxiliary retention are optimized.

---

> ### Author Rebuttal · Authors · 2026-03-30
>
> We sincerely thank Reviewer 4rjh for the thorough and constructive review. We are glad that the reviewer finds our 1D-to-2D paradigm shift effective. We address each concern below.
>
> > **W1.** The layout policy is heavily optimized for long-context QA tasks. It is unclear how well this salience allocation strategy transfers to more dynamic agentic workloads.
>
> To evaluate transferability beyond QA, we conduct **zero-shot evaluation on LOCOMO** [1] at the reviewer's suggestion.
>
> **Table R1.** Zero-shot evaluation on LOCOMO.
>
> | Budget | Mem-α | MemOCR |
> |---|---|---|
> | 1024 | 23.83 | **33.14** |
> | 256 | 23.05 | **31.71** |
> | 64 | 8.59 | **17.52** |
> | 16 | 3.12 | **15.63** |
>
> MemOCR outperforms Mem-α by ~9 points at full budget. Under extreme compression (16 tokens), MemOCR retains 15.63 while Mem-α drops to 3.12. These results demonstrate that MemOCR's layout policy transfers well across task types.
>
> > **W2.** The core conceptual leap is somewhat marginal. The methodology reads like a straightforward pipeline integration of DeepSeek-OCR and established text-based memory summarization paradigms.
>
> We respectfully disagree. Although both works involve OCR-related techniques, they differ fundamentally:
>
> 1. **DeepSeek-OCR is a passive OCR tool; MemOCR is an RL-driven memory agent.** DeepSeek-OCR simply renders existing text as an image — no learned layout policy, no adaptive information density, no budget-aware training.
> 2. **MemOCR actively learns to control visual layout** through RL, deciding what to emphasize (via typography), what to compress, and how to allocate the visual canvas.
>
> |  | DeepSeek-OCR | MemOCR |
> |---|---|---|
> | Key Innovation | New architecture for accurate OCR | Visual memory agent that learns to draft and utilize visual memory |
> | Core Behavior | Extract text from a given image | Decide what to emphasize, compress, and how to allocate visual canvas |
> | Base Model | Specialized architecture | Model-agnostic (Qwen2.5-VL in our experiments) |
> | Training | Supervised on OCR tasks | RL on long-horizon QA tasks |
>
> To empirically verify this distinction, we test whether a stronger OCR model can serve as a drop-in replacement. If MemOCR were merely text summarization + image rendering, swapping in a superior OCR reader should improve performance. We conduct **additional experiments** using DeepSeek-OCR [3] as the memory reader, with two different memory drafters:
>
> **Table R2.** Reader × Drafter comparison (10K context).
> |Reader|Drafter|Budget|HotpotQA|2Wiki|NQ|TriviaQA|Avg|
> |---|---|---|---|---|---|---|---|
> |MemOCR|MemOCR|1024|**84.8**|**72.2**|**61.8**|**79.6**|**74.6**|
> |DeepSeek-OCR|MemOCR|1024|71.0|66.7|52.9|54.7|61.3|
> |DeepSeek-OCR|Mem-α|1024|50.0|35.9|39.1|39.8|41.2|
> |MemOCR|MemOCR|16|**67.2**|**57.9**|**42.8**|**80.8**|**62.2**|
> |DeepSeek-OCR|MemOCR|16|55.6|53.0|37.8|39.3|46.4|
> |DeepSeek-OCR|Mem-α|16|35.2|25.0|33.8|35.2|32.3|
>
> Two findings:
> 1. Replacing the reader with stronger OCR (DeepSeek-OCR + MemOCR's memory) results in decreased performance, confirming that MemOCR's advantage comes from the jointly learned drafting-layout policy, not from the rendering step.
> 2. Holding the reader constant (DeepSeek-OCR), MemOCR's drafted memory consistently outperforms Mem-α's memory. This demonstrates that the learned drafting policy itself provides significant value beyond simple text memory + OCR module.
>
> > **W3.** The evaluation heavily relies on concatenated QA datasets... lacks evaluation on long-context benchmarks such as LOCOMO [1].
>
> Please see our response to W1 (Table R1). LOCOMO evaluation confirms that MemOCR generalizes to conversational memory settings without retraining.
>
> > **W4.** The paper lacks comparative analysis against the most recent baselines—LightMem, HyMem, SimpleMem.
>
> We thank the reviewer for suggesting these baselines. We reproduce **SimpleMem** [2] using the authors' official code:
>
> **Table R3.** Comparison with SimpleMem (10K/30K context).
>
> | Method | HotpotQA | NQ |
> |---|---|---|
> | SimpleMem | 73.4 / 67.2 | 38.3 / 37.5 |
> | MemOCR@1024 | **84.8 / 75.1** | **61.8 / 49.2** |
> | MemOCR@256 | **82.2 / 75.4** | **57.3 / 48.8** |
>
> SimpleMem uses GPT-4o as backbone and Qwen3-Embedding-0.6B for retrieval. Key observations are:
> 1. MemOCR outperforms SimpleMem on both benchmarks,
> 2. even MemOCR at 256 tokens maintains a clear advantage, validating the compression efficiency of visual memory.
>
> Due to time constraints, we have not yet reproduced LightMem and HyMem and will continue during the rebuttal period.
>
> [1] Evaluating Very Long-Term Conversational Memory of LLM Agents
>
> [2] SimpleMem: Efficient Lifelong Memory for LLM Agents
>
> [3] DeepSeek-OCR: Contexts Optical Compression

---

> > ### Author Rebuttal · Reviewer_4rjh · 2026-04-01
> >
> > Thank you for the thorough rebuttal. Q1-3 are adequately addressed. I will raise my score to Weak Accept and looking forward to more comparative results.

---

> > > ### Author Response · Authors · 2026-04-07
> > >
> > > We sincerely thank the Reviewer for acknowledging our rebuttal and raising the score. As promised, we have continued running comparative experiments during the discussion period and report the updated results below.
> > >
> > > **Table R3 (updated).** Comparison with SimpleMem and LightMem (10K / 30K context, 4 benchmarks).
> > >
> > > | Method | HotpotQA | 2Wiki | NQ | TriviaQA | Avg |
> > > |---|---|---|---|---|---|
> > > | SimpleMem (unlimited budget) | 73.4 / 67.2 | 52.3 / 41.4 | 38.3 / 37.5 | 57.8 / 62.5 | 55.5 / 52.2 |
> > > | LightMem (unlimited budget) | 64.1 / 66.4 | 59.4 / 50.8 | 50.0 / 46.1 | 42.2 / 38.3 | 53.9 / 50.4 |
> > > | MemOCR@1024 | **84.8** / 75.1 | **72.2 / 73.7** | **61.8 / 49.2** | 79.6 / **81.3** | **74.6 / 69.8** |
> > > | MemOCR@256 | 82.2 / **75.4** | 71.2 / 72.8 | 57.3 / 48.8 | **79.7** / 80.9 | 72.6 / 69.5 |
> > > | MemOCR@64 | 77.6 / 68.1 | 62.9 / 66.2 | 51.0 / 43.6 | 77.6 / 80.7 | 67.3 / 64.7 |
> > >
> > > Both SimpleMem (GPT-4o backbone + Qwen3-Embedding retriever) and LightMem are now included. MemOCR outperforms both baselines by a substantial margin across all four benchmarks: +19.1 / +17.6 avg over SimpleMem and +20.7 / +19.4 avg over LightMem at 1024 budget. Even at an extreme budget of 64 tokens, MemOCR still yields leading performance.
> > >
> > > We are currently working on reproducing HyMem but have encountered some technical issues with the official codebase. We will continue to resolve this and update if possible. If the Reviewer has any further suggestions, we welcome additional feedback.
> > >
> > > Thank you again for the constructive engagement and recognition of our work.

---

### Official Review · Reviewer_2EAB · 2026-03-11

**Soundness:** 3
**Presentation:** 3
**Significance:** 3
**Originality:** 3
**Overall Recommendation:** 4
**Confidence:** 4

**Summary:**

MemOCR replaces 1D textual memory with 2D visual memory for long-context LLM agents. The agent writes a structured Markdown memory (e.g., headings, emphasis), renders it as an image, and retrieves information through a VLM. Training uses GRPO-based reinforcement learning with three objectives to preserve answer quality under aggressive token budgets.

Experiments on HotpotQA, 2WikiMultiHopQA, NQ, and TriviaQA with contexts ranging from 10K to 100K tokens and budgets of 8–1024 tokens show that MemOCR surpasses RL-trained textual memory baselines (MemAgent, Mem-α) by about 7 points at full budget, while degrading substantially more gracefully under compression.

The central hypothesis—that visual layout enlarges the RL action space from discrete text selection to continuous visual saliency allocation—is intriguing.

**Compliance With Llm Reviewing Policy:**

Affirmed.

**Final Justification:**

My concerns have been adequately addressed.

**Key Questions For Authors:**

Q1. Token-budget fairness: How many text-equivalent tokens does one vision token encode? If you give MemAgent/Mem-α 3–4× more text tokens (matching the information capacity of the rendered image), does MemOCR still outperform? This single experiment would decisively clarify whether the gains come from layout or from the vision encoder's compression rate.

Q2. Layout as optimization landscape: Can you provide training convergence curves (MemOCR vs. MemAgent at matched RL steps) or performance at reduced training-set sizes? This would substantiate the claim that layout's continuous action space yields genuine optimization benefits beyond a different final representation.

**Limitations:**

yes

**Strengths And Weaknesses:**

S1. Creative memory formulation with a clear optimization interpretation. MemOCR represents memory as a layout-controlled 2D canvas, expanding the RL policy’s action space from discrete retain/discard decisions to continuous visual-saliency allocation (e.g., font size, position, emphasis). This enables soft demotion of secondary information rather than binary pruning. Layout ablations (Table 1) further confirm that visual hierarchy is critical for robustness under tight token budgets.

S2. Strong robustness under extreme compression. MemOCR retains 62.2% accuracy at 16 tokens, compared with 31.6% for MemAgent, despite both using RL-based extraction. Notably, MemOCR achieves comparable baseline performance at 8 tokens to what text-based methods require 64 tokens to reach, indicating roughly 8× token efficiency.

S3. Strong empirical analysis and reproducibility. The paper includes layout ablations, oracle-region injections, RL objective ablations, statistical tests, runtime analysis, failure-case studies, and detailed implementation information (anonymous code, hyperparameters, and hardware), providing above-average transparency and reproducibility.

W1. Token-budget fairness is not established. The paper treats vision tokens and text tokens as equivalent units, but a vision encoder likely packs significantly more information per token. Both MemAgent (RL via DAPO) and Mem-α (RL with QA reward) are already RL-trained for key-information extraction, yet trail MemOCR by ~7 points even at the generous 1024-token budget. Since both sides have comparable RL optimization, the residual gap likely reflects encoding-rate asymmetry rather than layout superiority. Without an information-equivalent comparison (e.g., giving text baselines 3–4× more tokens, or matching FLOPs/bytes), the core claim is confounded with the simpler explanation that vision encoders are better compressors.

W2. Scope-evidence mismatch and missing nearest-neighbor baselines. The paper claims "long-horizon agentic reasoning" but evaluates only on QA benchmarks — no planning, multi-turn tool use, or persistent-agent tasks. Equally, the related work cites AgentOCR, Glyph, and DeepSeek-OCR as OCR-based compression methods but never compares against them experimentally, leaving the contribution's novelty and magnitude relative to the closest literature unclear.

W3. Layout's optimization benefit is asserted but unverified. While the conceptual argument that layout provides a smoother optimization landscape is appealing, no training convergence curves, data-scaling comparisons, or sample-efficiency experiments are provided. Training cost is high (64×A100, 14 days, ~21k GPU-hours), and without evidence that layout accelerates convergence relative to text-memory RL, the claim remains speculative.

---

> ### Author Rebuttal · Authors · 2026-03-30
>
> We sincerely thank Reviewer 2EAB for the constructive opinions. We address each concern below with new experiments and analysis.
>
> > **W1 & Q1: Token-budget fairness**
>
> We fully acknowledge the reviewer's concern over token-budget fairness.
>
> To establish token-budget fairness, we conducted two additional comparisons:
> 1. comparing MemOCR@B vs. baselines@4B, to enhance fairness by granting baselines more context tokens;
> 2. summarization-based text memory denoising before truncation, to enhance fairness by increasing baseline's information density.
>
> > If you give MemAgent/Mem-α 3–4× more text tokens, does MemOCR still outperform?
>
> **Table R1.** Token-budget fairness comparison. Scores are avg performance on 4 benchmarks under 10K context.
>
> | Method@Budget | MemOCR | MemAgent | Mem-α |
> |---|---|---|---|
> | @256 vs @1024 | **72.6** | 67.8 | 47.3 |
> | @64 vs @256 | **67.3** | **67.3** | 40.3 |
> | @16 vs @64 | **62.2** | 50.7 | 25.2 |
>
> > a vision encoder likely packs significantly more information per token ... the residual gap likely reflects encoding-rate asymmetry rather than layout superiority.
>
> We agree with the reviewer's opinion that vision encoder seems to be efficient information compressor.
> To enhance comparison fairness in terms of information density, we conduct **additional experiments** using Qwen2.5-7B-Instruct to summarize text-based memories before truncation.
>
> **Table R2.** Comparison with text-summarization for memory denoising. Scores are avg performance on 4 benchmarks under 10K context.
>
> |Budget|MemOCR|MemAgent|
> |---|---|---|
> |1024|**74.6**|66.8|
> |256|**72.6**|67.2|
> |64|**67.3**|62.0|
> |16|**62.2**|58.5|
>
> Key observations from Table R1 & R2:
> 1. Even with 4× more text tokens, baseline methods cannot outperform MemOCR at extreme budgets. MemOCR's advantage grows sharply as the budget tightens, confirming the compression robustness comes from the adaptive information density.
> 2. Though denoising significantly increases the low-budget performance of baseline methods, MemOCR demonstrates superior performance gain across all memory budgets.
>
> > **W2: Scope-evidence mismatch and missing nearest-neighbor baselines**
>
> We address both parts:
>
> **1. Beyond-QA evaluation.** We conduct **additional zero-shot evaluation on LOCOMO** [1], a long-term conversational memory benchmark:
>
> **Table R3.** LOCOMO zero-shot comparison.
>
> | Budget | Mem-α | MemOCR |
> |---|---|---|
> | 1024 | 23.83 | **33.14** |
> | 256 | 23.05 | **31.71** |
> | 64 | 8.59 | **17.52** |
> | 16 | 3.12 | **15.63** |
>
> MemOCR outperforms Mem-α by ~9 pts at full budget and shows dramatically better compression robustness (15.63 vs. 3.12 at B=16), confirming that adaptive density transfers beyond QA.
>
> **2. Nearest-neighbor baseline.** We introduce **DeepSeek-OCR** [2] as the nearest-neighbor comparison. Both methods use the same memory images generated by MemOCR.
>
> **Table R4.** Comparison between different OCR models (10K context).
>
> | Memory Reader | Budget | HotpotQA | NQ |
> |---|---|---|---|
> | Qwen2.5-VL (ours) | 1024 | **84.8** | **61.8** |
> | DeepSeek-OCR | 1024 | 70.9 | 52.9 |
> | Qwen2.5-VL (ours) | 16 | **67.2** | **42.8** |
> | DeepSeek-OCR | 16 | 55.6 | 32.8 |
>
> Despite superior raw OCR capability, DeepSeek-OCR substantially underperforms Qwen2.5-VL, suggesting that reading visual memory depends on layout comprehension rather than OCR strength alone.
>
> > **W3 & Q2: Layout's optimization benefit / convergence curves**
>
> We want to politely clarify: **our paper does not claim that layout provides a smoother optimization landscape.**
> Our claim is that layout enables adaptive information density for better low-budget robustness. At the reviewer's suggestion, we have added one convergence analysis to better reveal the training dynamics of our method.
>
> We compare MemOCR and MemAgent training reward curves (both trained with exact-match reward on HotpotQA):
>
> **Table R5.** Training convergence comparison, where numbers are answer exact-match reward.
>
> | Steps | MemOCR | MemAgent |
> |---|---|---|
> | 200 | 0.669 | 0.687 |
> | 400 | 0.698 | 0.706 |
> | 600 | 0.734 | 0.729 |
> | 800 | 0.761 | 0.752 |
> | 1000 | 0.792 | 0.782 |
>
> Both methods converge steadily. MemAgent leads slightly in early steps but MemOCR overtakes after ~600 steps and maintains the advantage through convergence.
>
> [1] Evaluating Very Long-Term Conversational Memory of LLM Agents
>
> [2] DeepSeek-OCR: Contexts Optical Compression

---

> > ### Author Rebuttal · Reviewer_2EAB · 2026-04-02
> >
> > Thank you for the detailed rebuttal. My concerns have been adequately addressed. I will raise my score to Weak Accept.

---

> > > ### Author Response · Authors · 2026-04-07
> > >
> > > We sincerely thank the Reviewer for the positive feedback and for raising the score, and are glad that our rebuttal has adequately addressed your concerns.
> > >
> > > We will continue to refine the manuscript based on the suggestions received during the discussion period. Thank you again for the constructive review.

---

### Official Review · Reviewer_Ueft · 2026-03-11

**Soundness:** 3
**Presentation:** 3
**Significance:** 3
**Originality:** 3
**Overall Recommendation:** 4
**Confidence:** 3

**Summary:**

A broad theme assessed by this study is how large language model (LLM) agents can effectively manage extensive interaction histories and perform long-horizon reasoning under strict context window constraints. Existing text-based memory systems suffer from uniform information density, where token costs scale linearly regardless of semantic importance, leading to the eviction of crucial evidence under extreme budget constraints. To address this, the authors proceed to explore a central concept: adaptive information density via a 2D visual layout. The paper proposes MemOCR, a multimodal memory agent that drafts rich-text memory (e.g., Markdown) and renders it into a 2D image. By training the agent with budget-aware reinforcement learning (GRPO) across diverse compression levels, MemOCR learns to emphasize crucial information with prominent typography (e.g., large headers) while compressing auxiliary details. Extensive experiments on multi-hop and single-hop QA benchmarks demonstrate that MemOCR achieves an 8× token-efficiency gain and exhibits remarkable robustness (graceful degradation) under extreme token budgets (e.g., 16 visual tokens) compared to strong text-based baselines.

**Compliance With Llm Reviewing Policy:**

Affirmed.

**Key Questions For Authors:**

- Have you evaluated the generated visual memory images (trained on Qwen2.5-VL) using other state-of-the-art vision-language models (e.g., GPT-4o, Claude 3.5 Sonnet)? Do other models naturally "attend" to the layout emphasis in the same way under aggressive downsampling?

- How does MemOCR perform beyond information-seeking QA? Could you provide a preliminary discussion or toy experiment on standard agentic benchmarks (e.g., WebShop, ALFWorld) to demonstrate if the learned drafting policy generalizes?

**Limitations:**

The authors have adequately discussed the limitations in Appendix B and E. The core limitation is: The agent may hallucinate facts if the visual encoder misinterprets blurred characters under extreme compression.

**Strengths And Weaknesses:**

Strengths:

- The results are highly convincing. MemOCR not only matches or slightly outperforms baselines given sufficient budgets but demonstrates exceptional robustness under extreme compression. The "8× token-efficiency gain" is a significant contribution to deploying agents in resource-constrained environments.

- The budget-aware RL framework is elegantly designed. It effectively prevents the shortcut policy of generating uniform, medium-sized text and successfully enforces the desired adaptive information density.

- The paper is well-structured and provides deep insights into why the method works. The oracle injection experiments (Fig. 5), ablation studies (Table 2), and detailed failure mode analysis (Appendix E) greatly strengthen the paper's scientific rigor.

Weaknesses:

- As frankly acknowledged in the Failure Mode B (Appendix E), when the rich-text memory grows too long, the fixed rendering canvas forces global font scaling. Under tight budgets, this turns the entire image (even headers) into unreadable pixel noise, leading to catastrophic failure.

- The method's success relies heavily on Qwen2.5-VL's capability to read blurry or pixelated text. If the underlying vision model lacks robust low-resolution OCR capabilities, the performance might degrade sharply.

- General Capabilities. Could you evaluate the final MemOCR checkpoint on a few standard general benchmarks (e.g., MMLU, MT-Bench) to quantify the "alignment tax"? Does the agent refuse to answer non-QA prompts or consistently reply with bizarre Markdown formats?

- Rendering Aspect Ratio. How exactly does Chromium handle the viewport width/height when rendering the Markdown? Have you tested formatting the memory into a multi-column layout or a fixed square aspect ratio? How does extreme stretching/squashing affect Qwen-VL's OCR performance?

- Lacks an ablation study on RL data. Please add ablation experiments on RL data selection and its impact.

---

> ### Author Rebuttal · Authors · 2026-03-31
>
> We sincerely thank Reviewer Ueft for the positive assessment and address each concern below.
>
> > **W1: Catastrophic failure when rich-text memory grows too long**
>
> We fully agree this is a known failure mode (documented in Appendix E). In practice, we enforce a `Max rich-text memory tokens` quota of 2048 tokens (Table 5), which restricts memory length and limits this failure. When the visual budget is sufficient (e.g., B=1024 in Table 1), the agent consistently produces correct answers.
>
> > **W2: Reliance on Qwen2.5-VL's OCR capability**
>
> We agree that basic OCR ability is a prerequisite. However, we hypothesize that stronger OCR does not necessarily yield better visual memory comprehension, as reading visual memory requires understanding layout hierarchy and learning multi-hop reasoning rather than just recognizing blurry text.
>
> To verify this, we conduct **additional experiments** replacing memory reader with DeepSeek-OCR [1], which demonstrates superior OCR over the Qwen2.5-VL series:
>
> **Table R1.** Comparison between different OCR models (10K context).
>
> | Memory Reader | Budget | HotpotQA | NQ |
> |---|---|---|---|
> | MemOCR | 1024 | **84.8** | **61.8** |
> | DeepSeek-OCR | 1024 | 70.9 | 52.9 |
> | MemOCR | 16 | **67.2** | **42.8** |
> | DeepSeek-OCR | 16 | 55.6 | 32.8 |
>
> Despite stronger raw OCR, DeepSeek-OCR substantially underperforms.
> Surprisingly, the comparison under blurred situation (16-token results) suggests that better OCR does not reduce the performance gaps at limited resolution.
> These results confirm that the ability to exploit visual memory is **not reducible to OCR strength**.
>
> > **W3: Alignment tax (MMLU, MT-Bench)**
>
> We conduct **new evaluation** on MMLU to check the alignment tax:
>
> **Table R2.** Alignment tax evaluation.
>
> | Model | MMLU |
> |---|---|
> | Qwen2.5-VL-7B-Instruct | 72.5 |
> | MemOCR | 71.1 |
>
> The alignment tax is only ~1 point, indicating negligible degradation. This is expected since our RL training focuses on memory drafting/reading behaviors without broadly fine-tuning general language capabilities.
>
> > **W4: Rendering aspect ratio**
>
> We use a HTML template with fixed font size for each heading level and a fixed-width viewport for rendering. Markdown content flows top-to-bottom (as shown in Figure 7). At your suggestion, we conduct **additional evaluation** under (1) a fixed square aspect ratio and (2) extreme stretching to a flat 1:10 image.
>
> **Table R3.** Ablation study on the visual aspect ratio. Scores are avg over 4 benchmarks.
>
> |Aspect Ratio|10K|30K|100K|
> |---|---|---|---|
> |Fixed width (original)|**74.6**|69.8|66.6|
> |Square|74.6|**70.9**|67.0|
> |1:10 stretching|72.6|69.2|**68.3**|
>
> All three aspect ratios yield comparable performance, with less than 2 points difference across all context lengths. This confirms that MemOCR's layout policy is robust to rendering aspect ratio variations.
>
> > **W5: RL data ablation**
>
> We **train a new variant** using NQ (single-hop) instead of HotpotQA (multi-hop) for 1k steps and evaluate on both multi-/single-hop benchmarks.
>
> **Table R4.** RL training data ablation (10K context).
>
> | Training Data | Budget | HotpotQA | NQ |
> |---|---|---|---|
> | / | 1024 | 67.2 | 45.3 |
> | / | 16 | 33.6 | 27.3 |
> | NQ | 1024 | 53.9 | **66.1** |
> | NQ | 16 | 32.8 | **45.3** |
> | HotpotQA | 1024 | **84.8** | 61.8 |
> | HotpotQA | 16 | **67.2** | 42.8 |
>
> Training on HotpotQA (multi-hop) yields substantially better performance across both benchmarks, while NQ-trained MemOCR still maintains reasonable low-budget performance (45.3 on NQ@16). This suggests that multi-hop data teaches a richer drafting policy that transfers to simpler tasks.
>
> > **Q1: Cross-model transfer**
>
> We conduct **new cross-model transfer experiments** using GPT-4o to read memory images generated by MemOCR.
>
> **Table R5.** GPT-4o reading MemOCR's visual memory on HotpotQA (10K/30K/100K context).
>
> | Memory Source | 10K | 30K | 100K |
> |---|---|---|---|
> | MemOCR's Memory | **87.7** | **83.5** | **82.9** |
> | MemOCR w/o Layout | 84.4 | 81.5 | 80.1 |
> | Mem-α's Memory | 65.6 | 57.0 | 63.3 |
> | w/o Memory | 53.9 | 49.4 | 47.2 |
>
> GPT-4o with MemOCR's memory surpasses even Qwen2.5-VL (87.7 vs. 84.8), and the layout variant consistently outperforms the w/o Layout variant, confirming that both the visual saliency cues and the layout-based hierarchy transfer across VLMs.
>
> > **Q2: Beyond QA — agentic benchmarks**
>
> We conduct **zero-shot evaluation on LOCOMO** [2], a long-term conversational memory benchmark.
>
> **Table R6.** LOCOMO zero-shot (no retraining).
>
> | Budget | Mem-α | MemOCR |
> |---|---|---|
> | 1024 | 23.83 | **33.14** |
> | 256 | 23.05 | **31.71** |
> | 64 | 8.59 | **17.52** |
> | 16 | 3.12 | **15.63** |
>
> MemOCR outperforms Mem-α by ~9 pts across all budgets and retains 15.63% at B=16 where Mem-α drops to 3.12%, demonstrating that MemOCR's adaptive density transfers beyond QA to conversational memory.
>
> [1] DeepSeek-OCR: Contexts Optical Compression
>
> [2] Evaluating Very Long-Term Conversational Memory of LLM Agents

---

### Decision · Program_Chairs · 2026-04-30

**Decision:**

Accept (regular)

**Comment:**

This paper introduces MemOCR, which replaces text-based agent memory with rendered visual memory to achieve adaptive information density under tight token budgets. Reviewers initially raised concerns about token-budget fairness, missing comparisons with VTC-related baselines, and limited evaluation beyond QA tasks; three of four raised their scores to Weak Accept after the authors provided targeted experiments on retrieval-based baselines, cross-model transfer, LOCOMO evaluation, and text summarization comparisons, while one reviewer maintained Reject over evaluation fairness and VTC positioning. The final version should incorporate the rebuttal experiments into the main paper and more explicitly situate the contribution relative to prior Visual-Text Compression work in the introduction.